# Striatal fast-spiking interneurons selectively modulate circuit output and are required for habitual behavior

**Justin K O'Hare[1,2], Haofang Li[3], Namsoo Kim[3], Erin Gaidis[3], Kristen Ade[1,2], Jeff Beck[1], Henry Yin[3], Nicole Calakos[1,2]\***

[1]Department of Neurobiology, Duke University Medical Center, Durham, United States; [2]Department of Neurology, Duke University Medical Center, Durham, United States; [3]Department of Psychology and Neuroscience, Duke University, Durham, United States

**Abstract** Habit formation is a behavioral adaptation that automates routine actions. Habitual behavior correlates with broad reconfigurations of dorsolateral striatal (DLS) circuit properties that increase gain and shift pathway timing. The mechanism(s) for these circuit adaptations are unknown and could be responsible for habitual behavior. Here we find that a single class of interneuron, fast-spiking interneurons (FSIs), modulates all of these habit-predictive properties. Consistent with a role in habits, FSIs are more excitable in habitual mice compared to goal-directed and acute chemogenetic inhibition of FSIs in DLS prevents the expression of habitual lever pressing. *In vivo* recordings further reveal a previously unappreciated selective modulation of SPNs based on their firing patterns; FSIs inhibit most SPNs but paradoxically promote the activity of a subset displaying high fractions of gamma-frequency spiking. These results establish a microcircuit mechanism for habits and provide a new example of how interneurons mediate experience-dependent behavior.
DOI: https://doi.org/10.7554/eLife.26231.001

**\*For correspondence:**
nicole.calakos@duke.edu

**Competing interests:** The authors declare that no competing interests exist.

## Introduction

Habit formation is an adaptive behavioral response to frequent and positively reinforcing experiences. Once established, habits allow routine actions to be triggered by external cues. This automation frees cognitive resources that would otherwise process action-outcome relationships underlying goal-directed behavior. The dorsolateral region of the striatum has been heavily implicated in the formation and expression of habits through lesion and inactivation studies (*Yin et al., 2004*; *Yin et al., 2006*), *in vivo* recordings (*Tang et al., 2007*; *Jog et al., 1999*), and changes in synaptic strength (*Shan et al., 2015*). More recently, properties of the dorsolateral striatum (DLS) input-output transformation of afferent activity to striatal projection neuron firing were found to predict the extent of habitual behavior in individual animals (*O'Hare et al., 2016*). Despite these observations, the cellular microcircuit mechanisms driving habitual behavior have not been identified.

DLS output arises from striatal projection neurons (SPNs), which comprise ~95% of striatal neurons and project to either the direct (dSPNs) or indirect (iSPNs) basal ganglia pathways. The properties of evoked SPN firing *ex vivo* linearly predict behavior across the goal-directed to habitual spectrum in an operant lever pressing task (*O'Hare et al., 2016*). Specifically, habitual responding correlates with larger evoked responses in both the direct and indirect pathways as well as a shorter latency to fire of dSPNs relative to iSPNs. To identify a microcircuit mechanism for habitual behavior, we manipulated the striatal microcircuitry to identify local circuit elements that modulated these habit-predictive SPN firing properties (*Figure 1A,B*).

**eLife digest** From biting fingernails to the daily commute, habits are sets of actions that can be completed almost without thinking and that are difficult to change or stop. Behavioral neuroscientists refer to habits as "stimulus-response" behaviors, and know that forming a new habit requires a region deep within the brain called the dorsolateral striatum. Indeed, in this region, the outgoing neurons – which make up 95% of the cells - respond differently to incoming signals in mice that have learned habits compared to non-habitual mice. However a question remained: what exactly was producing these differences?

O'Hare et al. have now found, unexpectedly, that the answer resides not in the 95% of outgoing neurons, but rather in a rare type of cell known as the fast-spiking interneuron. This cell is connected to many others and it appears to act like a conductor, orchestrating the previously identified changes in the output neurons. These findings were made using mice that had been trained to press a lever for a sugar pellet reward. Habit was measured by how long mice kept pressing even if they had just been allowed to eat their fill of pellets and the test lever was no longer dispensing pellets. Habitual mice continue to press the lever in this circumstance, while other mice do not.

O'Hare et al. found that inactivating the "conductor" cell made the output neurons respond in the opposite way to how they normally respond in habitual mice. Further experiments showed that fast-spiking interneurons were also more easily activated in habitual mice. To test whether this putative "conductor" cell was necessary for habitual behaviors, a technique known as chemogenetics was used to turn down its activity in habitual mice. Indeed, reducing activity in the conductor cell blocked the habitual behavior.

While some habits are a helpful and economical way to get through daily life, habits are also thought to be corrupted in a number of diseases such as neurodegenerative diseases, addictions and compulsions. Identifying this specific, yet rare, cell as a critical part of maintaining habits points out a new target to consider for therapies. Further work is needed before such treatments might become available to treat habit-related disorders; though O'Hare et al. are now taking steps in this direction by trying to work out how the fast-spiking interneuron changes its own activity when a habit is formed.

DOI: https://doi.org/10.7554/eLife.26231.002

Glutamatergic corticostriatal synapses express dopamine-dependent forms of long-lasting synaptic potentiation and depression (*Shen et al., 2008*), making these connections a fitting site for experience-dependent adaptation of striatal output. Although such plasticity accompanies changes in behavior, including the formation of habits (*Shan et al., 2015*; *Nazzaro et al., 2012*), it does not readily explain the finding that increased gain in the direct and indirect SPNs in habitual mice was balanced (*O'Hare et al., 2016*) since synaptic strengthening would occur separately on the two SPN classes through dichotomous mechanisms (*Shen et al., 2008*). In addition, within the DLS, habit-predictive SPN firing properties were distributed uniformly rather than in discrete subpopulations of SPNs (*O'Hare et al., 2016*). Because interneurons are often anatomically suited to tune SPN activity in a similarly broad manner through extensive axonal arbors (*Kawaguchi et al., 1995*; *Tepper et al., 2010*), we hypothesized that plasticity of striatal interneurons might underlie the habit-associated changes in striatal output.

Among the various interneuron types resident to the striatum (*Tepper et al., 2010*), parvalbumin-positive, fast-spiking interneurons (FSIs) provide the strongest source of local modulation, exerting strong, feedforward inhibition of SPNs via perisomatic GABAergic contacts onto virtually all SPNs (*Gittis et al., 2010*; *Koós and Tepper, 1999*; *Koos et al., 2004*; *Mallet, 2005*; *Taverna et al., 2007*; *Straub et al., 2016*; *Szydlowski et al., 2013*). Notably, FSIs are expressed in the dorsal striatum on a mediolateral gradient with the most residing in DLS (*Gerfen, 1985*). FSIs also preferentially innervate dSPNs relative to iSPNs (*Gittis et al., 2010*), suggesting a potential mechanism by which FSI-mediated inhibition could allow iSPNs to fire before dSPNs in response to coincident excitatory input. Based on these considerations, we hypothesized that FSIs might drive the habit-predictive circuit features through a disinhibitory mechanism that would promote SPN firing and a preferentially earlier activation of the direct pathway. Striatal FSI plasticity has been demonstrated through

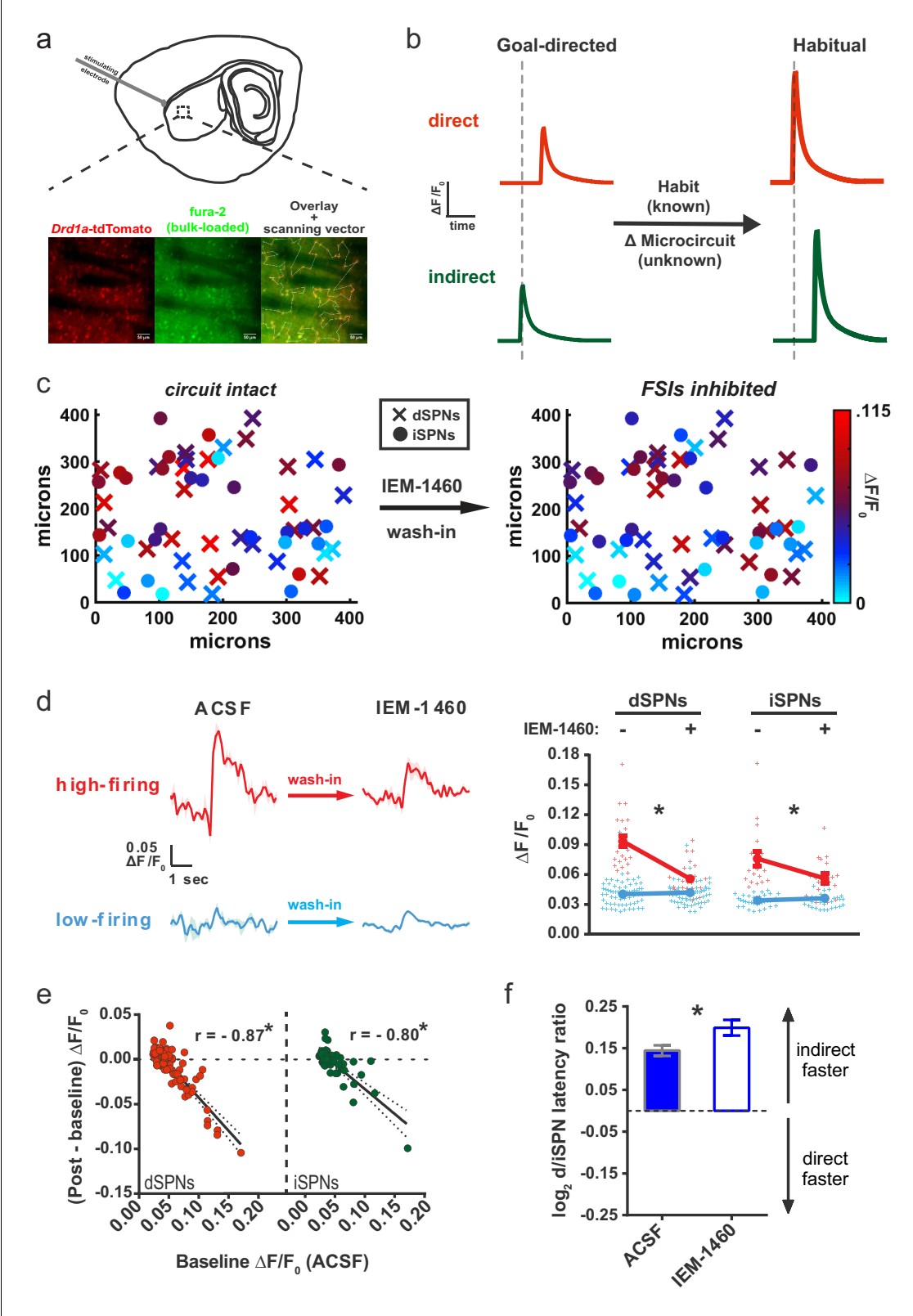

**Figure 1.** Striatal output reconfiguration following pharmacological inhibition of FSIs directly opposes substrates for habitual behavior. (**A**) Schematic of calcium imaging approach. Top: SPN activity was evoked by electrical stimulation of cortical afferent fibers in an acute parasaggital brain slice. Bottom: Evoked SPN firing was imaged in the direct and indirect pathways simultaneously using a transgenic direct pathway reporter mouse line (left), calcium indicator dye fura-2 (middle) and two-photon laser scanning microscopy (right, see scanning vector in overlay). (**B**) Experimental approach. Striatal

*Figure 1 continued on next page*

*Figure 1 continued*

microcircuitry was manipulated in tissue from untrained animals in order to reproduce the known circuit substrate for habitual behavior (described in *O'Hare et al., 2016*) and thereby identify a candidate microcircuit mechanism. (**C**) Representative heat maps of dSPN (x) and iSPN (•) calcium transient amplitudes before (left) and after (right) pharmacological inhibition of FSIs using IEM-1460 show a selective reduction in cells with the strongest (bright red) initial responses. (**D**) Left: Representative SPN calcium transient waveforms before and after wash-in of IEM-1460. SPNs were grouped into 'high-firing' (red) or 'low-firing' (blue) clusters based solely on their baseline response amplitudes using a Gaussian mixture model. SPNs with strong baseline responses (red, 'high firing') show weaker responses after wash-in whereas those with initially weak responses (blue, 'low firing') are unaffected. Right: Evoked calcium transient amplitudes for all imaged SPNs before (-) and after (+) wash-in of IEM-1460. For both cell types, high-firing SPNs showed decreased responses after IEM-1460 wash-in (dSPNs: $t(22) = 6.43$, $p=0.0000018$, n = 23 cells; iSPNs: $t(17) = 3.43$, $p=0.0032$, n = 18 cells) whereas low-firing SPNs did not (dSPNs: $p=0.24$, n = 64 cells; iSPNs: $p=0.21$, n = 34 cells). (**E**) Linear regression and correlational analyses show that the inhibitory effect of IEM-1460 on SPN responses (post – baseline difference) is a linear function of baseline response amplitudes for both dSPNs (red; $r(86) = -0.87$, $p=2.20\times10^{-28}$, n = 87 cells) and iSPNs (green; $r(51) = -0.80$, $p=1.59\times10^{-12}$, n = 52 cells). (**F**) Relative pathway timing, as measured by latency to peak detection, before and after inhibition of FSIs using IEM-1460. Indirect pathway activation precedes direct pathway activation by a greater margin after wash-in of IEM-1460 ($t(102) = 2.42$, $p=0.017$, n = 52 independent dSPN/iSPN pairs). *$p<0.05$. Dotted error bands indicate 95% confidence interval. Error bars indicate SEM. Effects of IEM-1460 on FSI and SPN spike probability are shown in *Figure 1—figure supplement 1*. Electrophysiological assessment of IEM-1460's effect on evoked multi-AP SPN responses is included in *Figure 1—figure supplement 2*. GMM parameters and calcium transient amplitude source data can be found in *Figure 1—source data 1*.

DOI: https://doi.org/10.7554/eLife.26231.003

The following source data and figure supplements are available for figure 1:

**Source data 1.** GMM parameters and source data for SPN calcium transient amplitudes (MATLAB).
DOI: https://doi.org/10.7554/eLife.26231.006
**Figure supplement 1.** IEM-1460 inhibits evoked FSI firing but does not affect SPN spike probability.
DOI: https://doi.org/10.7554/eLife.26231.004
**Figure supplement 2.** IEM-1460 selectively inhibits evoked multi-action potential SPN responses *ex vivo*.
DOI: https://doi.org/10.7554/eLife.26231.005

experimenter-induced activity and genetic manipulations (*Mathur et al., 2013*; *Winters et al., 2012*; *Orduz et al., 2013*; *Gittis et al., 2011a*), but it remains unknown whether dorsal striatal FSIs undergo plasticity normally in the context of experience-dependent adaptive behavior.

Using pharmacological and optogenetic manipulations, we found that striatal FSIs modulate the pathway-specific properties of DLS output that predict habitual behavior. Surprisingly though, silencing FSIs produced the opposite directionality for each habit-predictive circuit feature, suggesting that an increase, rather than decrease, in FSI activity might drive habitual behavior. Indeed, when FSI firing was evoked *ex vivo* by stimulation of cortical afferents, FSIs from habitual mice fired more readily than FSIs from goal-directed mice. To test the significance of this plasticity for the expression of habitual behavior, we acutely inhibited FSIs in DLS chemogenetically. Inhibiting FSIs in habit-trained mice blocked habit expression, but not lever-pressing per se, while identically-trained control subjects displayed robust habitual behavior. *In vivo* recordings revealed that the effects of FSI activity on striatal output appear to be more selective than previously appreciated. While FSIs exert the expected strongly inhibitory influence over DLS output, they also promote activity in a subset of SPNs that can be identified *a priori* based upon individual SPN firing patterns. Our results identify a mechanism for habit by which FSI strengthening reconfigures DLS output and promotes the expression of habitual behavior.

## Results

### Inhibiting fast-spiking interneurons drives a striatal circuit endophenotype opposite that of habitual behavior

To manipulate FSI activity, the calcium-permeable AMPA receptor (CP-AMPAR) antagonist IEM-1460, which predominantly weakens excitatory synaptic inputs onto FSIs in striatum (*Gittis et al., 2011b*), was used. Striatal FSIs express AMPARs lacking the GluA2 subunit, rendering them permeable to calcium (*Hollmann et al., 1991*), whereas SPNs do not typically express CP-AMPARs. Consistent with this difference in AMPAR subunit expression, IEM-1460 does not affect excitatory synaptic currents in SPNs but strongly decreases excitatory transmission onto FSIs (*Gittis et al., 2011b*). Cell-attached FSI recordings before and after exposure to IEM-1460 (50 μM) confirmed the drug's

efficacy to reduce synaptically-evoked AP firing in our acute parasagittal DLS preparation (*Figure 1—figure supplement 1*). To first approximate how FSIs modulate the habit-predictive properties of evoked striatal output, the same *ex vivo* population calcium imaging approach that identified the behavior-predictive properties (*O'Hare et al., 2016*) was used on tissue prepared from untrained animals (*Figure 1A,B*). Firing responses evoked by electrical activation of cortical afferents were measured in dozens of pathway-defined SPNs of both types simultaneously using the calcium indicator dye fura-2AM, the *Drd1a*-tdTomato (*Ade et al., 2011*) reporter, and vector-mode two-photon laser scanning microscopy (2PLSM) (*Figure 1A*; see Materials and methods and *O'Hare et al., 2016*). Action potential responses were detected by cross-correlation analysis with a template waveform that was obtained from single-action potential responses during simultaneous cell-attached electrophysiological recordings for each SPN subtype (see Materials and methods). Contamination of dSPN and iSPN datasets by interneurons was minimized by selection criteria and monitoring datasets for outliers (See Materials and methods for further details).

Firing properties of SPNs were compared within-cell before and after wash-in of IEM-1460. IEM-1460 decreased the amplitude of evoked calcium transients in both dSPNs (t(86) = 3.42, p=0.001, n = 87) and iSPNs (t(51) = 2.11, p=0.040, n = 52). IEM-1460 also changed the relative latency to fire between direct and indirect pathway SPNs by increasing the pre-existing bias in relative pathway timing whereby iSPNs tend to respond to cortical excitation more quickly than dSPNs (*Figure 1F*) (mean absolute latency values for dSPNs: 144.03 ± 7.08 ms ACSF, 154.33 ± 7.92 ms IEM-1460, N = 87; iSPNs: 130.31 ± 7.87 ms ACSF, 134.43 ± 8.89 ms IEM-1460, N = 52).

Upon closer inspection, the decrease in calcium transient amplitude seen at the population level appeared to be dominated by the subset of SPNs with larger baseline responses (for example, see brightest red cells before wash-in in *Figure 1C*). To determine whether there was selectivity for IEM-1460's effects on SPNs with large basal responses, calcium transient amplitude was used as a feature to classify SPNs as having large or small evoked calcium transients prior to drug wash-in. Rather than specifying an arbitrary cutoff value for the transient amplitude, we used an unsupervised clustering algorithm known as a Gaussian mixture model (GMM) to separate SPNs into two clusters. Based on calibration data in this preparation demonstrating the relationship between calcium transient amplitude and number of action potentials (*O'Hare et al., 2016*), the GMM separated SPNs into clusters corresponding to multi-action potential (larger transients; 'high-firing') and single-action potential (smaller transients; 'low-firing') responses (*Figure 1D*). Compared to the use of a physiologically-based 0.05 $\Delta F/F_0$ cutoff value, the unbiased GMM classification was in 90.5% agreement. According to this pre-IEM-1460 categorization, low-firing SPNs were unaffected whereas calcium transient amplitudes of high-firing SPNs were significantly reduced by IEM-1460 (*Figure 1D*).

This selective relationship was also borne out by examining the relationship between basal calcium transient amplitude and the magnitude of IEM-1460 effect. Consistent with a selective inhibition of multi-action potential responses, basal calcium transient amplitudes linearly predicted the inhibitory effect of IEM-1460 in both SPN subtypes (*Figure 1E*). Moreover, IEM-1460 did not affect spike probability in either SPN subtype (*Figure 1—figure supplement 1*). These pharmacological experiments in acute brain slices indicate that IEM-1460 promotes an indirect pathway timing advantage and selectively diminishes multi-action potential evoked SPN responses.

Because the within-cell experimental design of measuring effects before and after IEM-1460 application did not exclude the possibility that changes in calcium signals occurred during the 20 min wash-in period independently of IEM-1460, we performed a separate across-group study. Brain slices were incubated with either IEM-1460 or vehicle prior to and during imaging. Group mean calcium transient amplitudes were lower in IEM-1460 relative to vehicle in both dSPNs (vehicle: 0.043 ± 0.0011 $\Delta F/F_0$, N = 202 cells; IEM-1460: 0.037 ± 0.0021 $\Delta F/F_0$, N = 72 cells; t(272) = 2.62, p=0.0093) and iSPNs (vehicle: 0.040 ± 0.0014 $\Delta F/F_0$, N = 143 cells; IEM-1460: 0.033 ± 0.0011 $\Delta F/F_0$, N = 56 cells; t(197) = 2.93, p=0.0038) and IEM-1460-treated slices showed a preference for faster indirect pathway activation relative to vehicle-treated slices (t(197) = 3.83, p=1.41×$10^{-7}$, N = 143 and 56 independent dSPN/iSPN pairs). These results are generally consistent with findings from within-cell pre-post measurements.

To further test whether IEM-1460 selectively inhibited multi-spike SPN responses using methodology that did not involve inferring action potentials through calcium imaging, we used conventional electrophysiological methods to record cortically-evoked SPN firing in cell-attached mode. Brief single-pulse electrical stimuli (300–600 μs) were calibrated to elicit a stable multi-action potential

response in SPNs prior to taking a baseline measurement. Responses to the same stimulus were then recorded after wash-in of IEM-1460 or vehicle. Consistent with the calcium imaging results, IEM-1460 decreased evoked SPN firing (t(7) = 2.37, p=0.029, n = 8) while vehicle had no significant effect (p=0.76, n = 8). Moreover, the same selectivity for modulating multi-action potential responses was observed in that the magnitude of IEM-1460's effect correlated with the size of baseline responses and there was no effect on single-action potential responses (*Figure 1—figure supplement 2*). This result confirms that IEM-1460, which inhibits FSI firing (*Figure 1—figure supplement 1*), selectively reduces multi-action potential SPN responses to afferent stimulation as suggested by calcium imaging experiments (*Figure 1D,E*).

Altogether, this series of experiments identifies a pharmacological agent that potently inhibits FSI activity and modulates all of the habit-predictive SPN firing properties. These results were surprising for two reasons. First, rather than a blockade of FSI activity causing disinhibition of SPNs as we had hypothesized, we found that when FSI activity was reduced, SPN activity was also reduced. This result suggests that FSI activity is capable of promoting, rather than inhibiting, SPN activity at least in the acute brain slice preparation. Secondly, although IEM-1460 strikingly affected the same features of DLS output that predict the expression of habitual behavior (calcium transient amplitude in both pathways and relative pathway timing) (*O'Hare et al., 2016*), the directionality of these effects was opposite in all measures. Therefore, these results revise the overall hypothesis to involve a *gain*, rather than loss, of FSI activity as a candidate mechanism for habitual behavior.

## Parvalbumin-positive interneurons selectively promote multi-action potential SPN responses to cortical excitation *ex vivo*

While IEM-1460 has been shown to have selective effects on the firing of FSIs in striatum, its effect of inhibiting AMPAR-mediated excitatory postsynaptic currents (EPSCs) in cholinergic interneurons (CINs) (*Gittis et al., 2011b*) leaves open the possibility that CINs might contribute to our observed IEM-1460 effects. To isolate the effects of FSIs, the light-activated hyperpolarizing proton pump Archaerhodopsin-3 fused to green fluorescent protein (Arch-GFP) was Cre-dependently expressed in parvalbumin (PV)-expressing cells. *Pvalb*-Cre mice were crossed to a line which Cre-dependently expressed Arch-GFP (See Materials and methods). Control experiments showed that, as predicted, 532 nm light drove outward currents in FSIs but not SPNs (*Figure 2—figure supplement 1*). Additionally, Arch expressed in PV+ cells (PV-Arch) abolished high-frequency firing of FSIs in response to somatic current injection (*Figure 2—figure supplement 1*) and had no effect on SPN firing in the same recording configuration (*Figure 2—figure supplement 1*).

To examine the contribution of FSI activity to SPN firing, cortically-evoked SPN action potentials were recorded in cell-attached mode, as in the cell-attached IEM-1460 experiments, while nearby PV+ interneurons (~0.5 mm radius from recorded SPN) were silenced in alternating trials with 532 nm light exposure (*Figure 2A*). In this configuration, PV-Arch effectively blocked evoked FSI firing (*Figure 2B*). We found that optical inhibition of PV+ interneurons reliably decreased evoked SPN firing (*Figure 2C*, left and middle panels). Given that IEM-1460 selectively reduced the probability of multi-action potential SPN responses, we examined whether optical inhibition of PV+ neurons had a similar selectivity. Analysis of SPN responses by trial (paired consecutive laser OFF/ON sweeps), rather than by cell, indicated that single-action potential events and failures were unaffected when FSIs were silenced (*Figure 2C*, right panel). Moreover, a single-exponential fit of all trial-by-trial data showed a selective contribution of FSIs to multi-spike SPN responses (*Figure 2C*, right panel). Consistent with the IEM-1460 results in 2PLSM calcium imaging (*Figure 1D–E*) and cell-attached recording (*Figure 1—figure supplement 2*) experiments, this optogenetic result indicates that FSIs promote multi-action potential SPN responses to cortical excitation in the brain slice and that the effects of IEM-1460 on striatal output occur primarily through a reduction of striatal FSI activity.

## FSIs undergo long-lasting plasticity to become strengthened with habit formation

While results thus far show that FSIs appear capable of specifically modulating habit-predictive properties of striatal output, we next examined whether FSI activity was different as a result of experience. We measured FSI synaptic and cellular electrophysiological properties in DLS brain slices

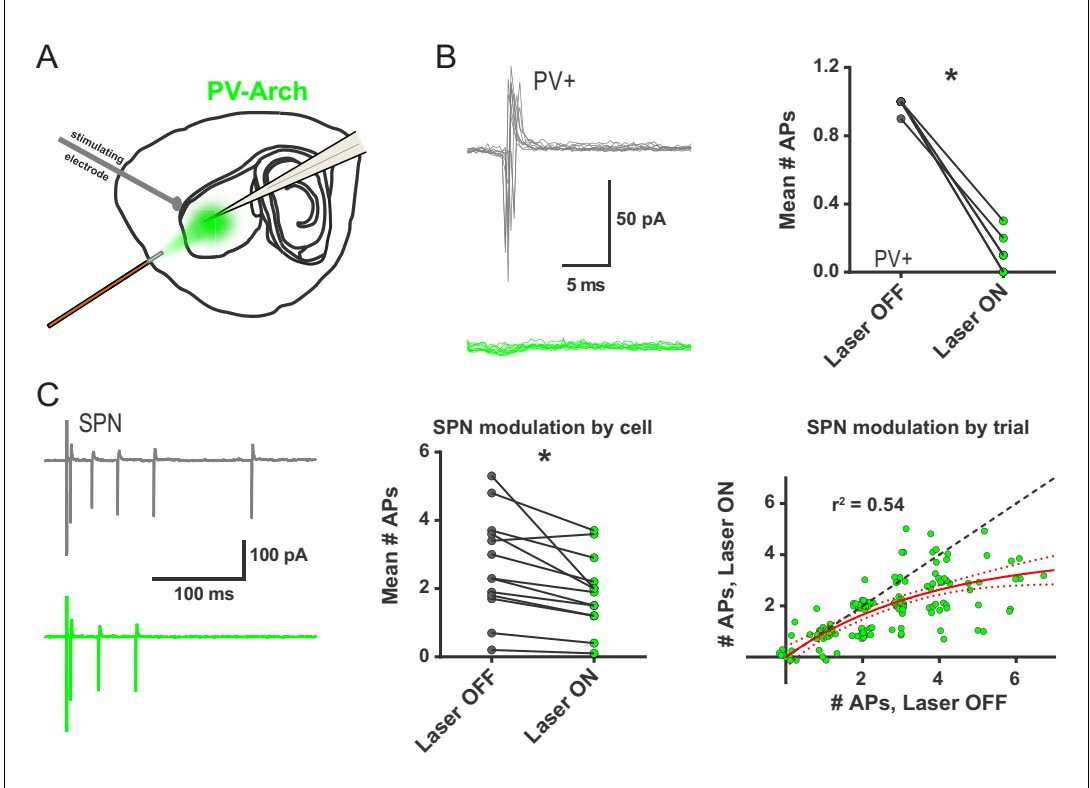

**Figure 2.** *Ex vivo* optogenetic inhibition of FSIs selectively reduces evoked multi-action potential SPN responses. (**A**) Experimental setup to record cortically-evoked action potentials in cell-attached mode with interleaved optogenetic inhibition of striatal FSIs. (**B**) Example traces (left) and mean number of APs (right) for evoked FSI firing with laser off (grey) and on (green). 532 nm light strongly inhibits evoked FSI firing (t(5) = 15.54, p=0.000020, n = 6 cells). (**C**) Evoked SPN action potential firing with interleaved optical inhibition of striatal FSIs. Left: Example traces showing consecutive sweeps of evoked multi-AP SPN firing with laser off (grey) and on (green). Middle: Mean number of evoked SPN APs with laser off (grey) and on (green). Inhibition of striatal FSIs caused SPNs to fire fewer action potentials (t(12) = 3.33, p=0.0060, n = 13 cells). Right: Data in middle plot shown as individual laser ON-OFF paired trials instead of by cell. Black dashed line denotes hypothetical regression line if laser had no effect. Data were jittered in x and y with Gaussian N(0, 0.15) to visualize overlapping points. Single exponential fit consistent with specific laser effect on multi-AP SPN responses (τ=13.78, r²(127)=0.54, n = 130 paired trials from 13 cells). *p<0.05. Dotted error bands indicate 95% confidence interval. Cell type specificity of Arch expression is shown in *Figure 2—figure supplement 1*.

DOI: https://doi.org/10.7554/eLife.26231.007

The following figure supplement is available for figure 2:

**Figure supplement 1.** 532 nm light selectively inhibits FSIs in PV-Arch mice *ex vivo*.

DOI: https://doi.org/10.7554/eLife.26231.008

prepared from habitual and goal-directed mice. *Pvalb*-Cre mice were bilaterally injected with AAV5-Ef1a-DIO-eYFP in the DLS to label PV+ interneurons and subsequently trained on an operant task in which they learned to press a lever for sucrose pellet rewards. Lever presses were reinforced on a random interval (RI) schedule to induce habit formation (*Dickinson et al., 1983*; *Hilário et al., 2007*) or on an abbreviated random ratio (RRshort) schedule to produce goal-directed behavior (*O'Hare et al., 2016*) (*Figure 3—figure supplement 1*). Habit was measured by evaluating the sensitivity of the learned lever press behavior to devaluation of the sucrose pellet reward. Goal-directed performance is known to be highly sensitive to outcome devaluation whereas habitual performance is less sensitive (*Dickinson et al., 1983*; *Hilário et al., 2007*; *Dickinson, 1985*). The sucrose pellet reward was devalued by inducing sensory-specific satiety. Specifically, mice were pre-fed with the reward pellets or, as a control for general satiety-related behavioral changes, identically-sized normal grain pellets. On separate but consecutive days, mice were alternately pre-fed 1.3 g of either the sucrose pellet reward (devalued condition) or the grain-only pellet (non-devalued condition), counterbalancing which pre-feed condition was tested first. Lever press rates were then measured during brief 3 min probe tests without reinforcement. Habitual behavior was quantified in individual mice as

the $\log_2$ ratio of the devalued versus non-devalued lever press rates (normalized devalued lever press rate; $NDLP_r$). RI-trained mice with an $NDLP_r \geq 0$, that is, insensitive to outcome devaluation, were considered to be habitual. RRshort-trained mice with an $NDLP_r < 0$ were considered to be goal-directed (*Figure 3—figure supplement 1*, shaded regions). Mice not meeting either inclusion criterion were not used for the electrophysiological studies.

We first examined whether excitatory synaptic transmission onto FSIs was altered with habit formation. Spontaneous EPSCs (sEPSCs) were recorded in the presence of the $GABA_A$ receptor antagonist picrotoxin (50 µM). No difference was detected in sEPSC frequency or amplitude between goal-directed and habitual FSIs (*Figure 3A*). Additionally, paired-pulse ratios of evoked EPSCs measured at a 50 ms inter-stimulus interval were similar between groups (*Figure 3B*). During these recordings, we also did not observe any group differences in a number of passive membrane properties (*Figure 3—figure supplement 1*).

Rather than changes in synaptic strength, we instead found robust differences in FSI firing responses to somatic current injection. FSIs from habitual mice displayed higher firing rates compared to FSIs from goal-directed mice (*Figure 3C*). Action potential kinetics did not appear to explain these group differences in firing rates as action potential waveforms were not appreciably different between groups (*Figure 3—figure supplement 1*). However, the duration over which firing could be sustained markedly differed between the two behavioral groups (*Figure 3D*). The majority of FSIs from goal-directed mice were unable to maintain high-frequency firing for the entire duration of the 500 ms current injection (<250 ms of firing in 10/15 cells) whereas nearly all FSIs from habitual mice maintained such activity (>450 ms firing in 7/9 cells). Interestingly, the distribution of goal-directed FSI response durations was strongly bimodal whereas that of habitual FSI response durations was not (*Figure 3D*). The group difference in response durations explained the difference in firing rates between FSIs of habitual and goal-directed mice since, when firing rates were normalized to the duration of firing instead of duration of the current step, there was no longer a group difference in firing rate (*Figure 3E*).

Habitual behavior was associated with increased FSI firing in response to somatic current injection. However, it was afferent activation that initially revealed habit-predictive striatal output properties (*O'Hare et al., 2016*). Therefore, in order for FSI plasticity to alter striatal output, it must be sufficient to differentially drive FSI firing in response to similar coincident synaptic excitation. FSI firing was monitored in cell-attached mode in response to electrical stimulation of excitatory afferents. We found that FSIs of habitual mice fired more readily than those from mice with goal-directed behavior (*Figure 3F*). This habit-related difference in FSI excitability was not readily explained by other aspects of lever pressing performance including the total number of lever presses or rewards delivered over the course of training (*Figure 3—figure supplement 1*). We noted the apparent bimodal distribution of total rewards delivered for goal-directed subjects (p=0.013, Hartigans' dip test; *Figure 3—figure supplement 1*) and wondered if the number of rewards received by an animal was related to the similarly-distributed FSI response durations to current injection (*Figure 3D*). Instead, we found that response durations from both modes of the distribution were commonly found in FSIs from the same goal-directed mouse (for example, 494.7 and 180.9 ms). Together, these experiments show that FSIs undergo long-lasting, experience-dependent plasticity with habit formation and that this plasticity is sufficient to increase FSI firing.

## FSI activity is required for the expression of a learned habit

Since photo-inhibiting FSIs produces striatal output properties that directly oppose those seen in habit (*Figure 1*), we inhibited FSIs after habit training to determine the necessity of FSI activity for expression of habitual behavior. Mice underwent habit-training protocols in the operant lever press task and then, prior to testing the degree of habitual responding, FSIs were inhibited chemogenetically. We selected a chemogenetic approach to allow for continuous modulation of activity during the 3 min probe tests which measure habitual behavior. *Drd1a*-tdTomato::*Pvalb*-Cre mice were bilaterally injected in DLS with AAV vectors Cre-dependently encoding either the inhibitory hM4D chemogenetic receptor (*Armbruster et al., 2007*) (PV-hM4D) or eYFP (PV-eYFP) (*Figure 4A,B*). Both groups underwent the same habit-promoting RI reinforcement protocol and learned similarly (*Figure 4C*). For both the devalued and non-devalued conditions, after each pre-feeding period and thirty minutes prior to the outcome devaluation probe tests, the hM4D agonist clozapine N-oxide (CNO, 5 mg/kg) was delivered intraperitoneally (*Figure 4D*).

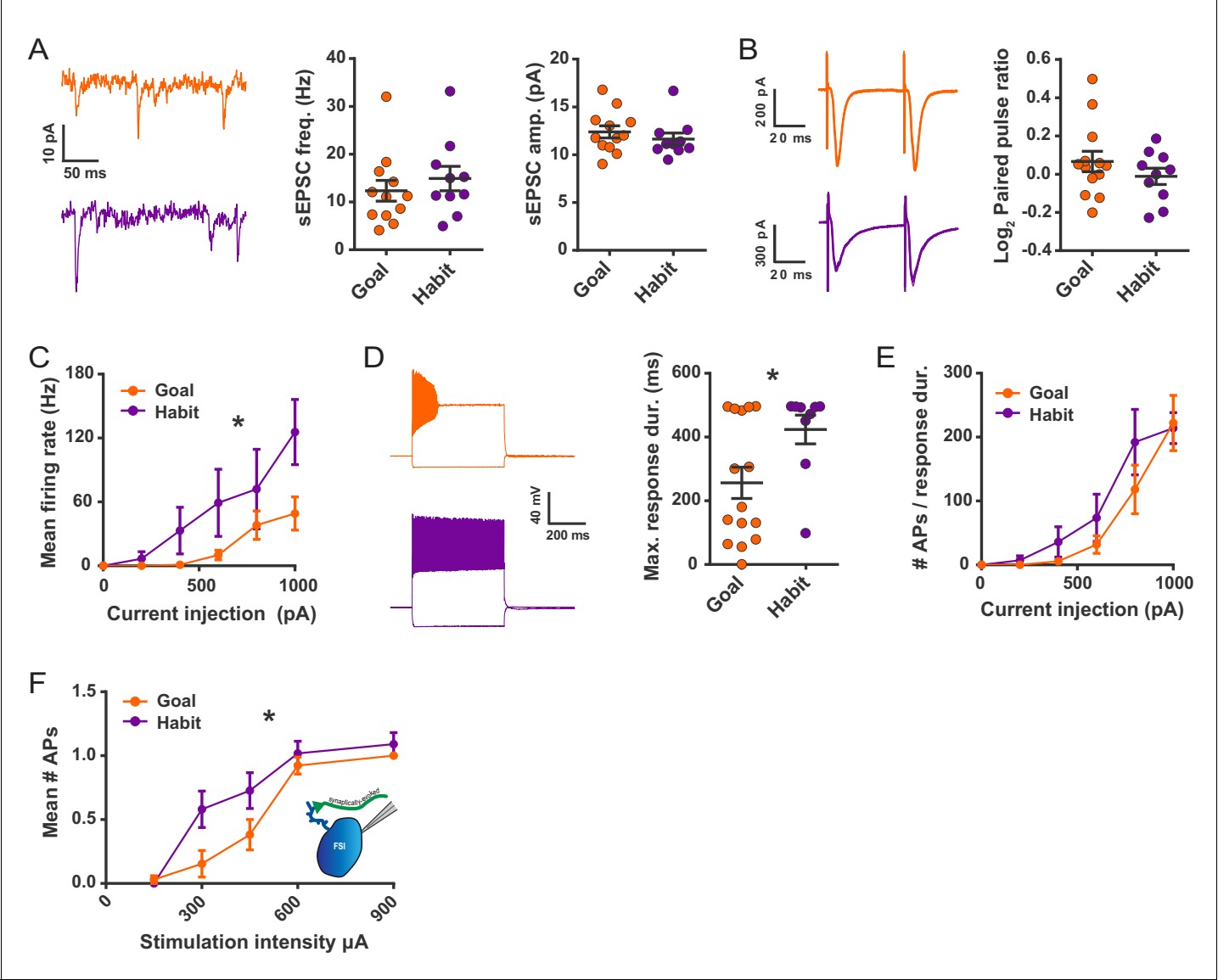

**Figure 3.** Habit formation enhances sustained high-frequency firing and cortically-evoked action potential firing in DLS FSIs *ex vivo*. (**A**) sEPSCs in FSIs of goal-directed (orange) and habitual (purple) mice. Left: Example sEPSC traces. No effect of training was found in sEPSC frequency (middle, p=0.45, n = 12 and 10 cells) or amplitude (right, p=0.42, n = 12 and 10 cells). (**B**) Paired-pulse measurements in FSIs of goal-directed and habitual mice. Left: Example traces showing FSI responses to paired single-pulse stimuli spaced 50 ms apart. Right: Habitual behavior was not associated with a change in paired pulse ratio relative to goal-directed behavior (p=0.29, n = 13 and 10 cells). (**C**) Input-output curve showing mean FSI firing rate in response to a series of increasing current steps. Habitual FSIs fired at an overall higher rate relative to goal-directed FSIs (F(1, 22) = 5.84, p=0.024, n = 15 and 9 cells). (**D**) FSI response durations, i.e. the time over which FSIs sustain firing. Left: Representative traces show that goal-directed FSIs often are unable to sustain firing for the duration of a 500 ms current step whereas habitual FSIs are typically able to do so. Right: Goal-directed FSIs are less-able to sustain firing than habitual FSIs (U = 34.5, p=0.049, n = 15 and 9 cells). Goal-directed response durations were bimodally distributed (p=0.020, Hartigans' dip test). (**E**) Firing rates as in (**C**) normalized to response duration. When accounting for response duration, no difference in firing rates is observed (p=0.25, n = 15 and 9 cells). (**F**) Input-output curve showing mean number of synaptically-evoked action potentials fired by goal-directed versus habitual FSIs in response to a series of increasingly strong single-pulse stimuli delivered to cortical afferent fibers. Responses recorded in cell-attached mode. Habitual FSIs fired more readily than goal-directed FSIs in response to afferent activation (F(1,22) = 4.77, p=0.040, n = 13 and 11 cells). *p<0.05. Data are represented as mean ± SEM. Additional behavioral and electrophysiological measures are included in *Figure 3—figure supplement 1*.

DOI: https://doi.org/10.7554/eLife.26231.009

The following figure supplement is available for figure 3:

**Figure supplement 1.** Electrophysiological properties of FSIs from habitual and goal-directed mice.

DOI: https://doi.org/10.7554/eLife.26231.010

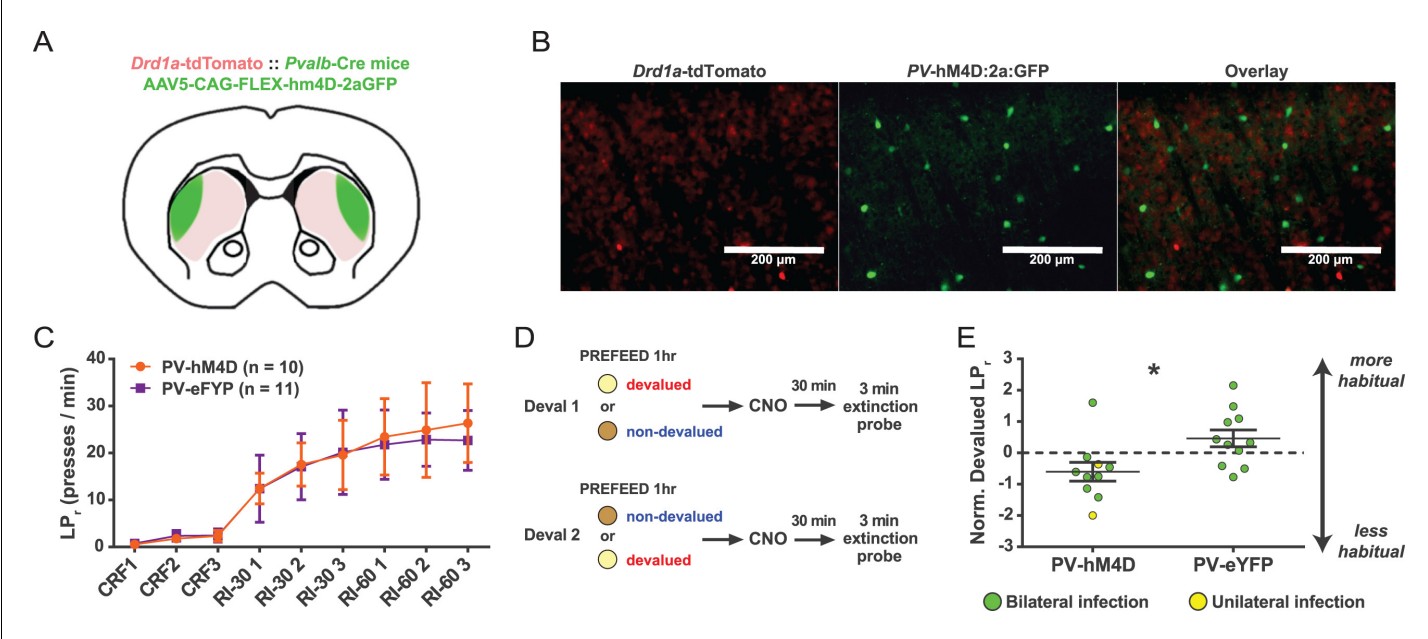

**Figure 4.** Acute chemogenetic inhibition of FSIs in dorsolateral striatum prevents expression of a learned lever pressing habit. (A) Diagram of coronal brain section showing tdTomato expression throughout striatum in dSPNs and expression of hM4D:2a:GFP construct in DLS. (B) Epifluorescent images of DLS showing tdTomato in dSPNs (left), GFP in PV+ cells (middle), and overlay (right). (C) Learning curves for hM4D and reporter construct-injected cohorts show that groups did not learn the task differently (p=0.70, n = 10 and 11 mice). (D) Experimental flow of devaluation testing to evaluate habit expression. Upon completion of multi-day training sessions, mice were pre-fed sucrose or grain pellets on alternating days, intraperitoneally administered CNO, and subjected to a 3 min extinction probe test 30 min later. Devalued (sucrose) and non-devalued (grain) lever press rates ($LP_r$) are compared ratiometrically using the normalized devalued $LP_r$ ($NDLP_r$) to assess habitual behavior: $NDLP_r = \log_2 \frac{devalued\ LP_r}{non-devalued\ LP_r}$. (E) Quantification of habit expression in individual subjects using $NDLP_r$. PV-hM4D mice showed less habit expression relative to PV-eYFP controls (t(19) = 2.66, p=0.016, n = 10 and 11 mice). *p<0.05. Data are represented as mean ± SEM. Effect of CNO on absolute $LP_r$ in the non-devalued condition is shown in *Figure 4—figure supplement 1*.

DOI: https://doi.org/10.7554/eLife.26231.011

The following figure supplement is available for figure 4:

**Figure supplement 1.** Chemogenetic inhibition of FSIs in dorsolateral striatum does not affect operant lever pressing in general.

DOI: https://doi.org/10.7554/eLife.26231.012

Chemogenetic inhibition of PV+ interneurons did not affect operant behavior in general, as evidenced by indistinguishable lever press rates between groups in the non-devalued (grain pellets) condition (*Figure 4—figure supplement 1*). In contrast, a comparison of sensitivity to outcome devaluation between groups revealed that habit expression was suppressed in PV-hM4D mice relative to PV-eYFP controls (*Figure 4E*). Mean $NDLP_r$ for RI-trained PV-eYFP control mice measured at 0.46 ± 0.27, indicating that control mice were insensitive to outcome devaluation, i.e. habitual. By contrast, PV-hM4D mice, which received the same RI training schedule and showed comparable rates of lever pressing (*Figure 4C*), displayed a mean $NDLP_r$ of −0.60 ± 0.30. A negative $NLDP_r$ indicates sensitivity to outcome devaluation, i.e. goal-directed responding. These findings show that acute suppression of FSI activity in DLS causes habit-trained subjects to behave as though they were goal-directed.

### FSIs exert an inhibitory net effect on striatal output *in vivo* while paradoxically promoting activity in subsets of high-bursting SPNs

To understand how chemogenetic suppression of FSI firing affects striatal activity *in vivo*, single unit recordings were performed in a cohort of *Drd1a*-tdTomato::*Pvalb*-Cre mice implanted in DLS with multi-electrode arrays and injected with the Cre-dependent hM4D inhibitory chemogenetic virus. Single units corresponding to both FSIs and SPNs were recorded in freely-moving mice (*Figure 5A–D*) for 30 min before intraperitoneal (i.p.) injection of CNO (5 mg/kg) or vehicle and during the

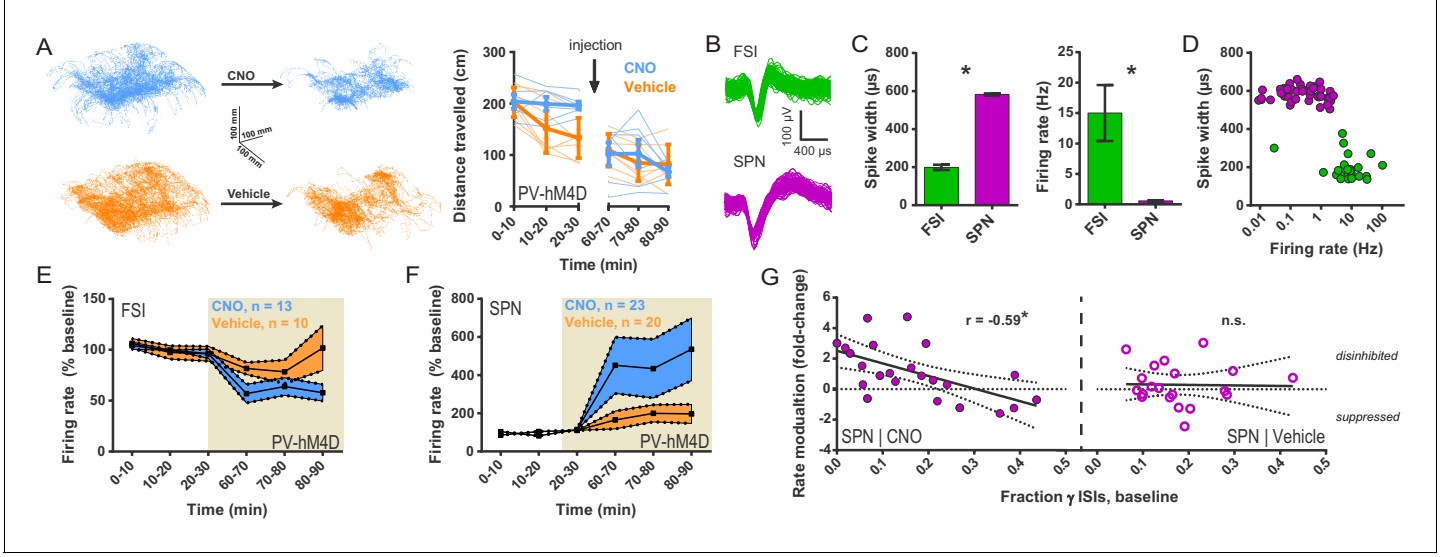

**Figure 5.** Chemogenetic inhibition of FSIs in DLS exerts a strongly disinhibitory net effect and selective excitatory effect on striatal output. (**A**) Locomotion before and after CNO or vehicle administration. Left: Example 3D traces showing head position during 30 min recordings before and after i.p. injection of CNO (blue) or vehicle (orange). Right: group-wise quantification of distance travelled shows that CNO- and vehicle-treated subjects did not respond differently to i.p. injections (p=0.16 for interaction of time and treatment, n = 6 and 7 mice). Subjects non-specifically decreased locomotor activity following the i.p. injection procedure (F(11,1) = 49.01, p=$2.27 \times 10^{-5}$, n = 6 and 7 mice). (**B**) Representative single-unit waveforms classified as FSIs (top, green) and SPNs (bottom, purple). (**C**) Waveform properties used for cell type classification. Left: FSI waveforms display a shorter spike width relative to those of SPNs (t(64) = 30.67, p=$5.53 \times 10^{-40}$, n = 23 FSIs and 43 SPNs). Right: FSIs display higher firing rates than SPNs (t(64) = 4.32, p=0.000056, n = 23 FSIs and 43 SPNs). (**D**) Classification of single units as FSIs (green) or SPNs (purple) by spike width and firing rate. (**E**) Time course showing FSI firing rates before (white background) and after (tan background) i.p. injection of CNO (blue) or vehicle (orange). CNO injection decreased FSI firing rate relative to vehicle (interaction between drug and time: F(5,105) = 2.51, p=0.034, n = 13 and 10 FSIs). (**F**) SPN responses to CNO or vehicle as in (**E**). CNO injection increased SPN firing rate relative to vehicle (interaction between drug and time: F(5,205) = 2.63, p=0.025, n = 23 and 20 SPNs). (**G**) Linear regression of fold-change ($log_2$ post/pre) in firing rate after CNO (left) or vehicle (right) injection against the baseline fraction of ISIs in the gamma frequency band. SPNs with higher fractions of gamma-frequency ISIs at baseline are more likely to decrease firing rate when FSIs are inhibited with CNO (r(22) = −0.59, p=0.0032, n = 23 cells) whereas vehicle caused no change in firing rate that could be predicted by baseline fraction of gamma ISIs (p=0.92, n = 20 cells). *p<0.05. Data are represented as mean ± SEM. Example units before and after CNO shown in *Figure 5—figure supplement 1*.

DOI: https://doi.org/10.7554/eLife.26231.013

The following figure supplement is available for figure 5:

**Figure supplement 1.** FSIs bidirectionally modulate firing rates as a function of baseline gamma spiking activity in individual SPNs.
DOI: https://doi.org/10.7554/eLife.26231.014

period of 30–60 min after injection. As expected for the inhibitory hM4D receptor, CNO significantly decreased FSI firing rates compared to vehicle-injected controls (CNO: 59.61 ± 8.08% baseline; vehicle: 86.89 ± 11.66% baseline) (*Figure 5E*). In line with previous *ex vivo* (*Koós and Tepper, 1999*; *Koos et al., 2004*) and *in vivo* (*Mallet, 2005*; *Gittis et al., 2011b*) studies, we further found that suppressing FSI activity caused an overall increase in SPN firing (i.e. disinhibitory effect) relative to vehicle (CNO: 472.00 ± 149.12%; vehicle: 188.02 ± 45.94%; *Figure 5F*).

In contrast to the straightforward effect of CNO on FSI activity, the effect of CNO injection on SPNs was far more variable. Post-CNO SPN firing rates ranged from 32.5% to 2511.1% of baseline (CV = 147%) with 26% of SPNs displaying negative modulation. In acute slice experiments, FSIs had displayed an unexpected and selective effect of promoting multi-action potential responses (*Figures 1D,E* and *2C*) but not otherwise affecting spike probability (*Figure 1—figure supplement 1*). To assess whether FSIs also promoted activity in identifiable subsets of SPNs *in vivo*, we analyzed the baseline firing patterns in single SPNs prior to CNO injection. SPN spiking was categorized into discrete frequency bands by deriving instantaneous firing rate from interspike intervals (ISIs) and was then normalized to total number of ISIs for each single unit. This analysis defined the fraction of ISIs corresponding to each frequency band for each SPN and was independent of local field potentials.

We found that the baseline (pre-CNO) fraction of ISIs falling within the highest rate frequency band, gamma-frequency (30–100 Hz), linearly predicted how firing rates in individual SPNs changed when FSI activity was suppressed (*Figure 5G*, left; see *Figure 5—figure supplement 1* for example units). That is, the higher the fraction of gamma-frequency spikes an SPN fired, the more likely it was to fire less when FSIs were chemogenetically inhibited. No such relationship was observed in response to vehicle (*Figure 5G*, right).

Since neurons with higher firing rates would be expected to have shorter ISIs in general, we examined the possibility that the fraction of gamma ISIs in SPNs might simply relate to mean firing rate. However, we found that the proportion of gamma-frequency ISIs was unrelated to mean firing rate in baseline single unit SPN recordings before either CNO or vehicle administration (pre-CNO: p=0.25, n = 23; pre-vehicle: p=0.28, n = 20). Additionally, we found that SPNs fire significantly more gamma-frequency spikes than expected by Poisson processes matched to firing rate (pre-CNO: t(44) = 5.76, p=$7.67\times10^{-7}$, n = 23 SPNs and rate-matched simulations; pre-vehicle: t(38) = 8.24, p=$5.59\times10^{-10}$, n = 20 SPNs and rate-matched simulations). Whereas baseline firing rates non-specifically predict fold change in firing rate after both CNO and vehicle injection (CNO: r(22) = −0.61, p=0.0022, n = 23; vehicle: r(19) = −0.45, p=0.045, n = 20), the excess probability of gamma-frequency ISIs (observed – expected) specifically predicts rate modulation after CNO (r(22) = −0.52, p=0.011, n = 23) but not vehicle (r(19) = 0.045, p=0.85, n = 20). Directly comparing correlation coefficients using Fisher r-to-z transformations showed that baseline firing rates did not predict SPN rate modulation by CNO better than by vehicle (z = −0.68, p=0.25) whereas baseline excess gamma specifically predicted modulation by CNO (z = −1.88, p=0.030). Therefore, gamma-frequency spiking represents a feature of interest in SPNs that predicts whether these output neurons will fire more or less as a consequence of reducing FSI activity.

These results demonstrate that FSIs modulate SPN activity in a more complicated manner than previously appreciated. While FSIs can have an overall strongly inhibitory effect *in vivo* on SPN firing as traditionally assumed, we also found evidence that they potentiate activity in a select population of SPNs that displays higher fractions of gamma-frequency spiking. This selective potentiation may be akin to a winner-take-all 'focusing' mechanism that increases the signal-to-noise ratio in corticostriatal transmission. According to such a mechanism, the subset of recruited SPNs would be facilitated while the less-relevant SPNs with low fractions of gamma spiking would be suppressed.

## Discussion

With the recent availability of tools to study specific, genetically-defined types of neurons, critical roles for interneurons in facilitating behavioral adaptations to experience are becoming increasingly apparent. In brain regions other than the striatum, interneuron activity appears to most commonly serve as a gate for the induction of long-lasting plasticity elsewhere in the local circuitry (*Kuhlman et al., 2013*; *Kvitsiani et al., 2013*; *Wolff et al., 2014*; *Yazaki-Sugiyama et al., 2009*). Although the potential for FSIs themselves to exhibit long-lasting activity-dependent plasticity is well-documented in acute brain slice experiments (*Mathur et al., 2013*; *Orduz et al., 2013*; *Hainmüller et al., 2014*; *Sarihi et al., 2012*; *Dehorter et al., 2015*), we are aware of only one report in which these interneurons were found to undergo experience-dependent plasticity and contribute to the expression of an adaptive behavior or memory (*Donato et al., 2013*). Here we provide the first such example for striatal interneurons. We find that FSIs are a site of adaptive plasticity that drives circuit and behavioral hallmarks of habit. The habit-associated changes in FSI excitability appear distinct (*Figure 3*, *Figure 3—figure supplement 1*) from previously reported plasticity processes which included activity-induced changes in FSI-SPN synapses selectively at direct pathway SPNs (*Mathur et al., 2013*) and changes in firing rate related to the modulation of afterhyperolarization currents by parvalbumin expression levels (*Orduz et al., 2013*). Further characterizing the plasticity mechanisms we find in habit represents an important area for future research as it may reveal a useful target for pharmacological modulation of FSI activity.

The approach we took to reveal the microcircuit mechanisms for habit was to identify a potential source for the broad local DLS circuit reorganizations of SPN firing properties that strongly correlate with habit (*Figure 1A,B*). To do this, we first examined how FSIs influenced striatal output using a pharmacological approach that inhibits excitatory synapses on striatal FSIs (and also CINs). In brain slices from untrained mice, IEM-1460 treatment showed striking specificity in that it modulated all of

the previously described (*O'Hare et al., 2016*) habit-predictive properties of evoked SPN firing *ex vivo*: gain of dSPN and iSPN responses (*Figure 1D,E*), and the relative timing of firing between dSPNs and iSPNs (*Figure 1F*). IEM-1460 also showed specificity in that it did not affect properties such as spike probability (*Figure 1—figure supplement 1*) that are not predictive of habit.

Unexpectedly, we found that the directionality by which FSIs modulated these properties was opposite to our original hypothesis: instead of the expected disinhibition of SPNs, silencing FSIs reduced SPN output (*Figure 1B–E*). FSI inhibition also altered the timing of direct and indirect pathway neuron firing in a direction that opposed the habit circuit signature (*Figure 1B,F*) and closely resembled previous observations in lever-press trained, goal-directed mice (*O'Hare et al., 2016*). This suggests that, in DLS, relative pathway timing is altered with habit formation but not with requisite goal-directed learning. Thus, the modest nature of the timing shift after pharmacological FSI blockade in untrained mice is likely due to a floor effect. Altogether, the observed effects of FSIs on SPNs lead to the prediction that an increase in FSI activity with habit formation would generate the evoked SPN properties that correlate with habit behavior (*Figure 1B*) (*O'Hare et al., 2016*). Accordingly, in habitual mice, we found that FSI firing was increased, and under the same cortical afferent stimulation conditions that evoke habit-predictive SPN firing properties (*Figure 3F*). This series of observations leads to a model of the striatal circuit basis for habitual behavior whereby habit formation is accompanied by a long-lasting increase in FSI excitability. In this setting, incoming cortical activity would be predicted to recruit more FSI activity that would in turn drive more firing of SPNs and shift their latencies such that direct pathway SPNs would tend to fire relatively sooner.

While anatomical and electrophysiological studies have long supported that striatal FSIs are critical for striatal circuit function (*Gittis et al., 2010*; *Koós and Tepper, 1999*; *Koos et al., 2004*; *Mallet, 2005*; *Taverna et al., 2007*; *Straub et al., 2016*; *Szydlowski et al., 2013*), an understanding of their specific behavioral contributions is much less developed. Prior *in vivo* studies have identified correlations of FSI activity with behaviors involving choice and reward-related actions (*Gage et al., 2010*; *Schmitzer-Torbert and Redish, 2008*), while more recent correlations of FSI activity with head movement velocity suggest another mechanism (*Kim et al., 2014*). In the present study, by chemogenetically inhibiting PV+ interneurons *in vivo*, we found that FSI activity in DLS is required for the expression of a learned habit (*Figure 4E*); an automated, reward-insensitive behavior quite different from behaviors previously studied. Previous pharmacological inactivation studies have demonstrated a role for DLS in habit expression (*Packard and McGaugh, 1996*; *Zapata et al., 2010*), indicating that general disruption of DLS activity also impairs established habitual behavior. Interestingly, in the present study, chemogenetic inhibition of FSI activity drove an overall increase in projection neuron activity (*Figure 5F*) which suggests that reducing FSI activity specifically may impair habit expression differently than a general inactivation of the circuitry.

While the disruption of habit by chemogenetically inhibiting FSIs supports a critical role for FSIs in this behavior, this experiment does not identify FSI plasticity as a mechanism for the expression of habit since artificially manipulating the activity of any cell that plays an otherwise critical role in the function of an implicated brain region might similarly disrupt behavior. Rather, in this study, a specific role for FSI plasticity as a mechanism for habit expression is indicated by the observations that these interneurons modulate those specific striatal output properties that correlate with habit (*Figure 1 and 2*) and show long-lasting changes in excitability after habit learning (*Figure 3D,F*).

Using opto- and chemo-genetic manipulations, we further found that FSIs, which are GABAergic, enhance activity in subsets of SPNs both in the acute slice and *in vivo*. Although it is unclear what if any relationship exists between the SPN subpopulations identified *ex vivo* versus *in vivo*, there exist multiple intriguing parallels. In the acute slice, only activity of those SPNs which displayed burst-like, multi-action potential responses to single-pulse stimuli ('high-firing' SPNs) was suppressed when FSIs were silenced (*Figures 1D,E* and *2C*). *In vivo*, the activity of SPNs showing the highest fractions of gamma-frequency spiking was suppressed, instead of disinhibited, when FSI activity was chemogenetically reduced (*Figure 5G*). In both cases, the SPNs were distinguished by a higher propensity for burst-like firing patterns. It was further notable that the fraction of SPNs negatively modulated by reduced FSI activity was similar in both preparations (29% *ex vivo* compared to 26% *in vivo*). Conversely, we also found that less-active SPNs were not significantly modulated in the slice (*Figures 1D,E* and *2C*) and SPNs with less gamma-frequency spiking were disinhibited *in vivo* when FSI activity was reduced (*Figure 5G*). This finding is reminiscent of a previous *in vivo* report that SPNs with weaker responses to cortical microstimulation displayed the most marked disinhibition

upon GABA$_A$ receptor blockade (*Mallet, 2005*). An important future direction will be to determine whether there are unique biological properties that distinguish the subset of SPNs whose activity is promoted, as opposed to inhibited, by FSIs.

Although an activity-*promoting* effect of GABAergic FSIs may appear counterintuitive, previous computational (*Humphries et al., 2009*) and biological (*Bracci and Panzeri, 2006*) studies describe such a phenomenon based in part on the 'up' and 'down' resting membrane potential states of SPNs that straddle the chloride reversal potential (E$_{Cl-}$). While a voltage-dependent excitatory effect of GABA would not necessarily affect spike probability due to a concurrent decrease in membrane resistance and the disparity between E$_{Cl-}$ and spike threshold, such an effect could boost the gluta-mate-driven depolarization of an SPN in its down state (*Humphries et al., 2009*; *Bracci and Panzeri, 2006*). Although disynaptic interneuron microcircuitry is a more common mechanism for disinhibitory effects of interneurons in other brain regions (*Wolff et al., 2014*; *Lovett-Barron et al., 2012*), some of our observations such as the influence of FSIs on SPN initial latency to fire (*Figure 1F*) are not con-sistent with the time delay necessitated by a disynaptic microcircuitry. For this reason, we instead favor a monosynaptic mechanism whereby properties of SPN resting membrane potential and firing patterns interact to yield activity-promoting effects of FSIs on SPN subsets.

Based on the previous observation that habit-predictive striatal output properties are relatively uniformly distributed when elicited by strong bulk stimulation of cortical afferents (*O'Hare et al., 2016*), it became apparent that habit-related adaptations of DLS broadly augment the propagation of cortical excitation into the basal ganglia. To confer specificity for certain actions, additional circuit dynamics would ostensibly be required. We hypothesized that such specificity could arise from the activation of subsets of task-specific cortical neuron projections that would in turn activate task-spe-cific SPNs (*Rothwell et al., 2015*; *Carelli and West, 1991*; *Gremel et al., 2016*). Indeed, recent evi-dence suggests that spatially-clustered SPN activity encodes information relevant to locomotor behavior (*Barbera et al., 2016*). In habits, one possible mechanism then is that task-specific cortical commands drive (*Smith et al., 2012*), or at least initiate (*Berke et al., 2004*), high-frequency firing in a cluster/subset of SPNs that would then be preferentially excited by FSIs. Additionally, in such a mechanism, feed-forward inhibition of less-active SPNs (*Mallet, 2005*) by FSIs might then serve as a selective filter to further enhance signal-to-noise ratio in corticostriatal transmission. One testable prediction of this model is that different behaviors would reveal different subsets of gamma-rich SPNs whose activity is promoted by FSIs.

Lastly, it is notable that FSIs are also implicated in some pathological settings associated with compulsive behavior. For example, fewer striatal FSIs, as determined by parvalbumin-immunopositiv-ity, have been observed in human brains from individuals with Tourette's syndrome (*Kalanithi et al., 2005*) and mouse brains in a model of OCD-like behavior (*Burguière et al., 2013*). OCD is highly comorbid in Tourette's syndrome (*Sheppard et al., 1999*) and disrupted habit learning has been implicated in pathological compulsivity in a variety of settings (*Graybiel, 2008*; *Everitt and Robbins, 2005*; *Gerdeman et al., 2003*). Interestingly, since both of the above studies defined FSIs by parval-bumin immunoreactivity, an intriguing alternative view of those results is that parvalbumin levels are below detection threshold but cell number is not necessarily reduced. Lower parvalbumin levels are associated with a hyperexcitable FSI phenotype (*Orduz et al., 2013*), which is akin to the direction of FSI plasticity we associate with habit in the present study. Thus, the finding of increased FSI excit-ability as a plasticity mechanism driving habitual responding also yields new insights to the potential mechanistic relatedness of habit and compulsion.

## Materials and methods

### Animals

All experiments were carried out under approved animal protocols in accordance with Duke Univer-sity Institutional Animal Care and Use Committee standards. Mice were 2–4 months of age, in C57Bl/6 genetic background, and were hemi-/heterozygous for all transgenes. *Drd1a*-tdTomato line 6 BAC transgenic mice were generated in our laboratory (RRID: IMSR_JAX:016204) (*Ade et al., 2011*). To optically inhibit PV+ interneurons, a mouse line expressing Cre under control of the *Parvalbumin* (*Pvalb*) promoter (RRID:IMSR_JAX:012358) was crossed to the Ai35D line from Jackson Laboratory which Cre-dependently expressed Arch3.0-GFP (RRID:IMSR_JAX:012735). To target PV

+ interneurons with Cre-dependent viral vectors, the *Drd1a*-tdTomato mouse line was crossed to the *Pvalb*-Cre line to produce experimental progeny hemizygous for *Drd1a*-tdTomato and heterozygous for *Pvalb*-Cre. For identification of PV+ neurons in 2PLSM calcium imaging experiments, the *Pvalb*-Cre mouse line was crossed to the Ai9 line (RRID:IMSR_JAX:007909) which Cre-dependently expressed tdTomato.

## Viral vectors

The *CAG*-FLEX-*rev*-hM4D:2a:GFP plasmid was provided by the Sternson laboratory at Janelia Farm (Addgene #52536). UNC Viral Vector Core packaged this plasmid into AAV 2/5 and also provided AAV2/5-EF1a-DIO-eYFP. All viral aliquots had titers above $1 \times 10^{12}$ particles/mL.

## Intracranial viral injections

Stereotaxic injections were carried out on 2–3 month old *Drd1a*-tdTomato::*Pvalb*-Cre mice under isoflurane anesthesia (4% induction, 0.5–1.0% maintenance). Meloxicam (2 mg/kg) was administered subcutaneously after anesthesia induction and prior to surgical procedures for postoperative pain relief. Small craniotomies were made over the injection sites and 1.0 µL virus was delivered bilaterally to dorsolateral striatum via a Nanoject II (Drummond Scientific) at a rate of 0.1 µL/min. The injection pipette was held in place for 5 min following injection and then slowly removed. Coordinates for all injections relative to bregma were as follows: A/P: +0.8 mm, M/L: ±2.7–2.8 mm, D/V: 3.2 mm. Mice were allowed a minimum of 14 days recovery before behavioral training. For experiments involving chemogenetic inhibition of FSIs specifically in DLS, mice showing no expression or poor targeting (misses were medial to DLS) were excluded from the study prior to behavioral analysis and data unblinding. Two AAV2/5-*CAG*-FLEX-*rev*-hM4D:2a:GFP-injected mice showed expression in only one hemisphere of DLS. These mice were included for behavioral analysis and behaved no differently from bilaterally-infected mice. We note that exclusion of these two subjects does not affect the statistical significance of the result.

## Lever press training

Prior to training, animals were restricted to 85–90% baseline weight to motivate learning. Lever presses were rewarded with sucrose-containing pellets (Bio-serv, F05684) and grain-only pellets (Bio-serv, F05934) were used as a sensory-specific control for satiety. Mice were trained in Med Associates operant chambers housed within light-resistant, sound-attenuating cabinets (ENV-022MD). Lever presses and food cup entries were recorded by Med-PC-IV software. During RR reinforcement, pellets were delivered every X times on average for an RR-X schedule. RI reinforcement gave a 10% probability of reward every X seconds for an RI-X schedule. Following random reinforcement training, subjects underwent devaluation testing to measure habitual behavior as previously described (*O'Hare et al., 2016*). When training schedule was a variable, experiments were performed with experimenter blind to training schedule.

For electrophysiological assessment of FSI properties, acute brain slices were prepared 0–24 hr after the final training session. Mice were excluded from analysis if they did not display the behavior that was expected based on training schedule. Specifically, mice that were trained to be habitual (random interval reinforcement) yet showed sensitivity to outcome devaluation (NDLP$_r$ <0) were excluded.

## Brain slice preparation

Animals were anesthetized using 2,2,2-tribromoethanol and transcardially perfused with ice-cold *N*-Methyl-D-glucamine (NMDG) solution (*Ting et al., 2014*). Brains were quickly removed and 300 µm thick parasaggital sections were cut in NMDG solution using a Leica VT1200S. For electrophysiological experiments, slices recovered at 32°C in NMDG solution for 10–12 min and were then transferred to room temperature HEPES-containing holding solution (*Ting et al., 2014*) where they remained for the rest of the experiment. Slices remained undisturbed in the HEPES holding solution for at least one hour prior to recording. For 2PLSM calcium imaging experiments, slices were allowed to recover for approximately 45 min in NMDG solution at room temperature. Slices were then transferred to room temperature HEPES holding solution (*Ting et al., 2014*) shortly before bulk-loading with fura-2, AM. Cutting and holding solutions were calibrated to 305 ± 1 mOsm/L. ACSF was calibrated to

305 ± 1 mOsm/L for 2PLSM calcium imaging and 315 ± 2 mOsm/L for electrophysiological recordings with internal solutions at 295 mOsm/L. Solutions were pH 7.3–7.4 and were carbogenated to saturation at all times.

## Drugs

For electrophysiological recordings, IEM-1460 was dissolved in deionized, distilled water at 100 mM and added to carbogenated ACSF for a final concentration of 50 µM. Picrotoxin was dissolved at 200 mM in DMSO and added to ACSF at 50 µM. For behavioral experiments, CNO was dissolved to 10 mg/mL in DMSO and diluted in sterile 0.9% saline solution to administer 5 mg/kg per subject with a maximum injection volume of 0.5 mL. For *in vivo* electrophysiological recordings, CNO and vehicle were administered on different days and in counterbalanced order.

## Electrophysiological recordings

Data were acquired using an Axopatch 200B amplifier (Molecular Devices) and a Digidata 1440A digitizer (Axon Instruments). Data were digitized at 10–20 kHz and low-pass filtered at 2 kHz. Borosilicate glass pipettes were pulled to 2–5 MΩ resistance. Slices were continuously perfused with carbogenated ACSF (124 mM NaCl, 4.5 mM KCl, 1 mM MgCl$_2$·6 H$_2$O, 26 mM NaHCO$_3$, 1.2 mM NaH$_2$PO$_4$, 10 mM glucose, 4 mM CaCl$_2$) at a temperature of 29–31°C.

### Current clamp experiments

Fast-spiking interneurons were identified by Cre-dependent fluorescence as well as their characteristically narrow action potential half-width. Current clamp (and cell-attached) recordings were carried out using a potassium methansulfonate-based internal solution (140 mM KMeSO$_4$, 7.5 mM NaCl, 10 mM NaCl, 10 mM HEPES, 0.2 mM EGTA, 4.2 mM ATP·Mg, 0.4 mM GTP·Na3).

### Voltage clamp experiments

Fast-spiking interneurons were identified by Cre-dependent fluorescence as well as previously reported ranges for input resistance and whole cell capacitance (*Gittis et al., 2010*). Voltage clamp recordings were carried out using a cesium methanesulfonate-based internal solution (120 mM CsOH, 120 mM MeSO$_4$, 15 mM CsCl, 8 mM NaCl, 10 mM TEA-Cl, 10 mM HEPES, 2 mM QX-314, 4 mM ATP·Mg, 0.3 mM GTP·Na3).

### Cell-attached experiments

Stimuli were delivered to cortical afferent fibers at the cortical side of the internal capsule (*Figure 2A*) using a bipolar stimulating electrode (FHC, CBARC75). Responses in SPNs and FSIs were recorded in cell-attached configuration with voltage clamped at 0 mV. Leak current was continuously monitored to detect partial break-ins. In the event of a partial membrane rupture, leak currents increased significantly due to the voltage at which the membrane patch was clamped. In these events, data were discarded. The same potassium methanesulfonate-based internal solution as in the current clamp experiments was used to enable break-in and cell type identification or further recordings after cell-attached experiments concluded. All stimuli were delivered with a 20 s inter-stimulus interval. For input-output experiments, 300 µs single-pulse stimuli were delivered with 5 sweeps per intensity, in order from weakest to strongest intensity, and cells were recorded at a consistent distance from the stimulating electrode (600–650 µm). For pre-post experiments with application of IEM-1460, 300–600 µs single-pulse stimuli were delivered to drive multi-action potential responses prior to drug wash-in. 10 sweeps were analyzed as baseline and another 10 sweeps, using the same stimulus parameters, following a 20 min wash-in period were analyzed to measure drug effect.

### In vitro optical inhibition of FSIs

532 nm light was delivered from a diode-pumped solid state laser (Opto Engine) coupled to a 300 µm core, 0.39 NA patch cable which terminated into a 2.5 mm ferrule (Thorlabs Inc.). The ferrule was submerged in the perfusion chamber and positioned with a micromanipulator to illuminate a ~0.5 mm radius around the tip of the recording pipet. Laser onset coincided with electrical stimulation of cortical afferents. Laser stimulation lasted 500 ms in whole cell current clamp experiments and 1 s when monitoring synaptically-evoked responses in cell-attached mode.

## *In vivo* single-unit recordings

Custom-made multi-electrode arrays were used for all recordings. The arrays consisted of fine-cut tungsten wires and a 6-cm-long silver grounding wire. Tungsten wires were 35 μm in diameter and 6 mm in length, arranged in a 4 × 4 configuration. The row spacing was 150 μm, and electrode spacing was 150 μm. All arrays were attached to the 16-channel Omnetics connector and fixed to the skull with dental acrylic. After hM4D viral injection into the dorsolateral striatum, the electrode arrays were lowered at the following stereotaxic coordinates in relation to bregma: 0.8 rostral, 2.75 lateral, and 2.6 mm below brain surface. Single-unit activity was recorded with miniaturized wireless headstages (Triangle BioSystems International) using the Cerebus data acquisition system (Blackrock Microsystems), as previously described (*Fan et al., 2011*). The chronically implanted electrode array was connected to a wireless transmitter cap (~3.8 g). During recording sessions, single units were selected using online sorting. Infrared reflective markers (6.35 mm diameter) were affixed to recording headstages to track mouse position as subjects moved freely on a raised platform. Marker position was monitored at 100 Hz sampling rate by eight Raptor-H Digital Cameras (MotionAnalysis Corp.). Before data analysis, the waveforms were sorted again using Offline Sorter (Plexon). Only single-unit activity with a clear separation from noise was used for the data analysis. In each case, a unit was only included if action potential amplitude was ≥5 times that of the noise band. FSIs and SPNs were classified on the basis of spike width and baseline firing rate (*Figure 5B–D*).

## 2PLSM calcium imaging of DLS output

Synaptically-evoked action potential firing was monitored in dozens of direct and indirect pathway SPNs simultaneously as previously described (*O'Hare et al., 2016*) in acute brain slices prepared from untrained *Drd1a*-tdTomato hemizygous mice (*Ade et al., 2011*) aged 2–4 months. Detailed methods are included below.

### Bulk-loading of fura-2, AM

Fura-2, AM (Life Technologies, F-1221) was dissolved in a solution of 20% pluronic acid F-127 (Sigma) in DMSO by vortexing and sonication. The solution was then filtered through a microcentrifuge tube. Slices were transferred to small loading chambers with room temperature ACSF + 2.5 mM probenecid (osmolality and pH readjusted to 305 ± 1 and 7.3–7.4). 1.1 μL fura-2, AM solution was slowly painted directly onto the striatum of each slice. Additional fura-2 AM solution was added as needed to reach a final DMSO concentration of 0.1% by volume. Slices were incubated in a dark environment for 1 hr at 32–33° with continuous carbogenation of the loading chambers. The prolonged 1 hr incubation was found to be necessary for satisfactory loading in acute slices prepared from adult and aging animals.

### Selecting field of view

After the incubation period, slices were moved to carbogenated HEPES holding solution. Slices remained in holding solution until used for an experiment, at which point they were moved to a recording chamber and continuously perfused with carbogenated ACSF (124 mM NaCl, 4.5 mM KCl, 1 mM MgCl$_2$·6 H$_2$O, 26 mM NaHCO$_3$, 1.2 mM NaH$_2$PO$_4$, 10 mM glucose, 4 mM CaCl$_2$) at a temperature of 29–31°C. To evoke SPN responses, cortical afferents were stimulated in bulk by a bipolar concentric electrode (FHC, CBARC75) placed at the dorsoanterior edge of the internal capsule (*Figure 1A*). A 410 × 410 μm field of view (FoV) was selected by following the cortical fibers along a diagonal ventroposterior path from the electrode at a distance of 600–650 μm. At this distance, SPN action potentials can be evoked without the cells being directly depolarized (*O'Hare et al., 2016*). Fura-2 and tdTomato (expressed in dSPNs) were excited simultaneously at 750 nm (fura-2 isosbestic wavelength) using a Ti: Sapphire laser (Chameleon Ultra 1, Coherent Inc.). Red and green photons were collected by separate photomultiplier tubes (PMTs) both above and below the microscope stage.

### Classifying regions of interest

Regions of interest (ROIs) showing red and green were classified as dSPNs whereas green-only ROIs were classified as iSPNs (*O'Hare et al., 2016*). The small percentage of green-only cells which would have been striatal interneurons was partially mitigated by ignoring abnormally large ROIs which were likely to be cholinergic interneurons. FSIs comprise approximately 1% of all striatal neurons

and are not present in the *Drd1a*-TdTomato labelled population (*Ade et al., 2011*). However, FSIs might be included in the 'putative iSPN' population. Empiric experiments using Pv-Cre x Ai9 reporter mice demonstrate that approximately 50% of FSIs are labelled with fura2-AM and pass data inclusion criteria in our experimental setting (see *Figure 6*). Therefore, we expect at most that approximately 1% of iSPN and none of dSPN data may represent FSIs. Because CIN and FSI firing properties are very different from SPNs, we reviewed our data sets for the presence of outliers using

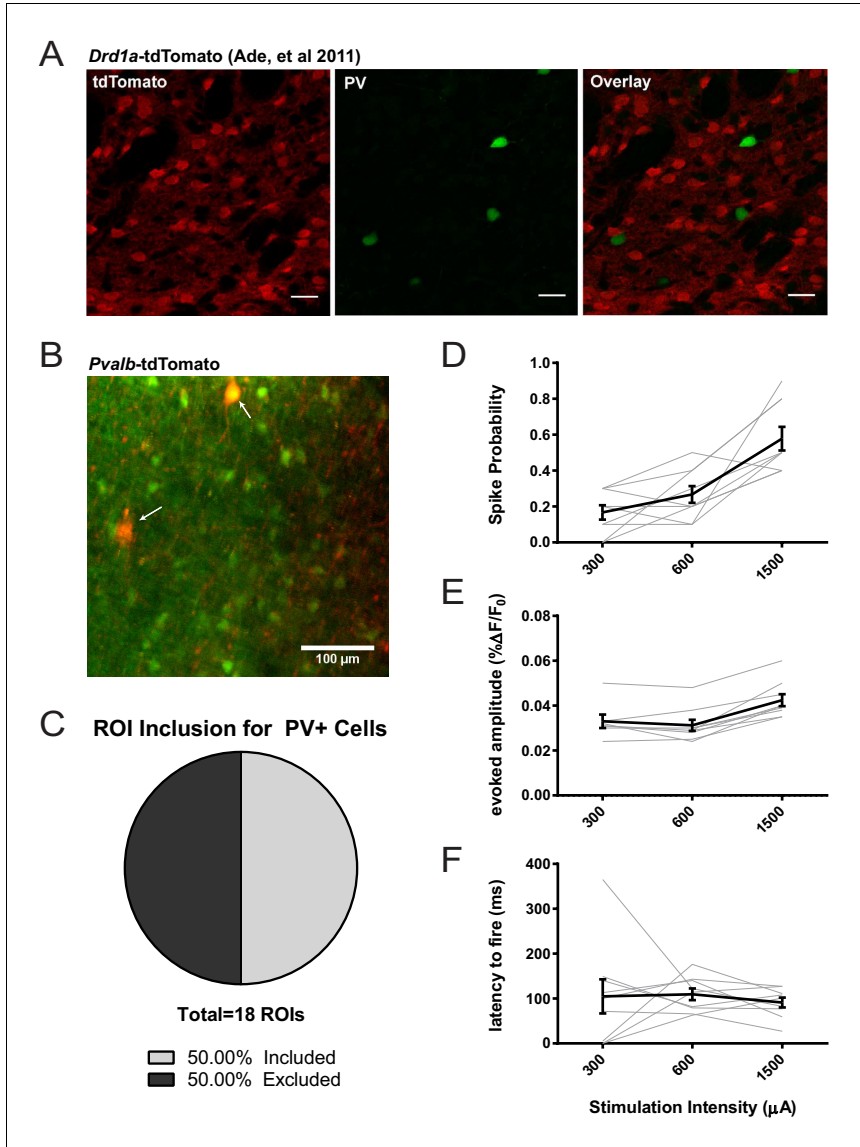

**Figure 6.** Assessing potential contribution of FSIs to SPN 2PLSM datasets. (**A**) Figure panel adapted from *Ade et al. (2011)* showing that the BAC transgenic *Drd1a*-tdTomato mouse that was used in 2PLSM experiments described in *Figure 1* does not express tdTomato in striatal PV+ neurons. tdTomato signal (left) was not detected in neurons staining positive for the parvalbumin antigen (middle) as shown in overlay (right). Scale bars represent 25 μm. (**B**) Field of view for 2PLSM calcium imaging of PV+ neurons in *Pvalb*-Cre/Ai9 mice which Cre-dependently express tdTomato. Fura-2-loaded PV+ neurons were identified by transgenic expression of tdTomato (indicated by white arrows) and overlapping fura-2 signal (green) (**C**) Pie chart showing that only 50% of transgenically-labelled PV+ neurons are sufficiently loaded with fura-2 and pass other ROI criteria to be included in 2PLSM calcium imaging analysis (n = 18 cells). (**D-F**) Firing properties of PV+ neurons, as a function of stimulation intensity at cortical afferent fibers, including spike probability (D), amplitude of evoked calcium transients (E), and latency to fire (F) (n = 9 cells).
DOI: https://doi.org/10.7554/eLife.26231.015

Robust regression and Outlier (ROUT) analysis (GraphPad Prism software) and a false discovery rate of Q = 1%. No outliers were detected in iSPN baseline amplitude, post-IEM-1460 amplitude, or latency modulation datasets of *Figure 1*.

## Population imaging of evoked SPN calcium transients

ROIs were manually selected in ImageJ until no ROIs remained in the FoV. A matrix of ROI centroid coordinates was then imported to PrairieView to generate a linescan vector. If temporal resolution for the linescan vector fell below the minimum of 12 Hz due to an overabundance of ROIs (and thus lengthy vector), they were removed in order of increasing light intensity in the green channel. Centroid coordinates were permanently attributed to each ROI in order to retain spatial information along with SPN subtype and firing properties. Fura-2 signal was measured at each ROI along the scan path at 770 nm in response to 300 μs, 300 μA single-pulse stimulation of cortical afferents. Images were acquired at a frequency of 12–15 Hz. A diffractive optical element was used to increase signal-to-noise (SLH-505D-0.23–785, Coherent Inc.) (*Watson et al., 2009*). Photons were collected by a 40X, 0.8 NA LUMPlanFL water immersion objective lens and Aplanat/Achromatic 1.4 NA oil immersion condenser (Olympus). Red and green photons were directed to dedicated PMTs by a 575 nm dichroic mirror. Images were acquired via PrarieView image acquisition software and were time-locked to stimuli by Trigger Sync (Bruker Corp.). Stimuli were delivered 10 times with a 20 s interstimulis interval. For each ROI, firing properties were calculated at the single-trial level before being averaged across trials.

## Data analysis

All experiments and data analyses were performed with experimenter blind to the experimental variable (e.g. viral construct, training schedule). *A priori* sample sizes were established based on power analyses. Data exclusion criteria and decisions were made prior to data unblinding.

### Cell-attached experiments

Action potentials were detected in cell-attached mode by cross-correlating data to a template waveform. Template waveforms were composite action potentials recorded in cell-attached mode from single neurons that were positively identified as the corresponding cell type in a subsequent whole cell current clamp recording. The dot product representing a perfect fit was obtained by cross-correlating the template peak to itself. If the dot product of the data and the template peak was equal or greater than 25% of this perfect fit, then an action potential was called by a peak detection algorithm (Mathworks, Inc.). Stimulus artifacts and rare spontaneous action potentials were excluded by only analyzing data from 1 to 100 ms (FSIs) or 1–600 ms (SPNs) after stimulus delivery. Due to the sharp FSI cell-attached waveform, electrical noise was matched to the FSI template peak in some recordings. To exclude these false calls, two additional exclusion criteria were added: action potentials were excluded (1) if their amplitudes were less than 10 standard deviations of the recording minus the stimulus artifact, i.e. electrical noise and (2) if their cross-correlation peak amplitudes were less than 25% of the maximum peak in a given sweep.

### Current-clamp experiments

Action potentials were detected by running a peak detection algorithm (Mathworks Inc.) on voltage velocity data with a peak threshold of $1 \times 10^4$ V/s and a minimum peak distance of 2 ms. Action potential onset and offset were defined at the intersections of the waveform with a sliding mean baseline voltage that constituted 10% of the length of the current injection. Action potential and after-hyperpolarization properties were measured up to the point when increasing current injection attenuated firing rate. Action potential half-width was defined as half the time between onset and peak voltage. Action potential amplitude was defined as the voltage difference between the sliding baseline and peak amplitude. AHP potential onset and offset were defined as the next two intersections with the sliding baseline after the action potential peak voltage. AHP amplitude was defined as the negative-most voltage between onset and offset and the AHP waveform was integrated over the sliding baseline for total voltage. AHP voltage measurements were converted to current using input resistance. Firing rates were measured in response to a series of increasing 500 ms current step amplitudes ranging from −0.4 to 2.0 nA in 200 pA intervals. Maximum response duration was

defined as the longest period of sustained firing observed during this series of current injections. Rheobase was determined by identifying the 200 pA interval in which the first action potential was fired and subsequently interrogating this interval with 500 ms current injections at 10 pA resolution. Subthreshold test pulses were used to determine passive membrane properties. Input resistance was calculated as $R_I = dV/I$. Whole cell capacitance was calculated by integrating the decay phase after current injection to measure discharged current and dividing by voltage of the current injection: $\int V_{decay}/IR_i$ (*Gittis et al., 2010*). Series resistance was calculated by fitting a standard double-exponential function to the decay transient and deriving the time constant $\tau = 1/\lambda_{fast}$ to find $\tau_{fast} = R_s \times C_{whole\ cell}$. Cells with $R_s$ >30 megaOhms were excluded from analysis.

## Voltage clamp experiments

Voltage clamp experiments assessing habit-related FSI physiology were carried out in the presence of picrotoxin (50 µM). Paired pulse ratio was calculated as $\log_2(EPSC_2/EPSC_1)$ for first and second EPSC amplitudes. Paired stimuli were delivered 50 ms apart. Spontaneous EPSCs were recorded at $V_m = -70$ mV at 5X gain for 5 min per cell. Automated event detection was performed using MiniAnalysis (Synaptosoft). To validate the use of PV-Arch, 532 nm light-induced currents were recorded in FSI and SPNs in the presence of gabazine (10 µM), AP5 (50 µM), and NBQX (50 µM) to block $GABA_A$, NMDA, and AMPA receptors, respectively.

## *In vivo* single-unit recordings

Single unit activity was sorted into frequency bins by converting interspike intervals to instantaneous firing rates. Frequency bands were defined as $\Delta$ = 0–4 Hz, $\theta$ = 4–8 Hz, $\alpha$ = 8–13 Hz, $\beta$ = 13–30 Hz, and $\gamma$ = 30–100 Hz. The fraction of ISIs falling in a particular frequency band was calculated relative to the total number of ISIs. To compare frequency band distributions of single unit records to rate-matched Poisson processes, for each single unit with N ISIs, N points were randomly drawn from a Poisson distribution with $\lambda$ set to the mean ISI (1/mean firing rate) for the corresponding single unit. This simulation was run 20 times per single unit. All 20 simulations were binned according to the described frequency band bounds and normalized counts were averaged across simulations. Since each simulated unit corresponded to a real recording with mean firing rate = $1/\lambda$, observed and simulated data were compared via multiple paired t-tests and Bonferroni-Sidak correction for multiple comparisons. For behavioral analysis, 3D tracking data were transformed into Cartesian coordinates (x, y and z) by the Cortex software (MotionAnalysis Corp.) to allow distance calculations.

## 2PLSM calcium imaging

Raw frames were corrected using a drift correction algorithm (*Li et al., 2008*) to control for minor fluctuations in X and Y. Baseline fluorescence was measured over a 2 s sliding window to calculate change in fluorescence over baseline ($\Delta F/F_0$). Action potentials were detected using a cross-correlation approach as described for current clamp and cell-attached recordings above. The template peak was generated by simultaneous calcium imaging and cell-attached electrophysiological experiments and represented a single action potential (*O'Hare et al., 2016*). Detected peaks possessed dot-products at least 50% that of a perfect fit (cross-correlating template to itself). Although dSPN and iSPN calcium transients are similar in these experimental conditions (*O'Hare et al., 2016*), separate template peaks corresponding to the SPN subtype classification of each ROI were used.

Additional inclusion criteria beyond the cross-correlation threshold were used at the level of event detection, ROI inclusion, and slice inclusion to maximize data quality and reliability. Detected events were included as evoked responses only if they occurred within 375 ms of stimulation- any other events were excluded from analysis. Additionally, a lockout window was set in the peak detection algorithm to ensure that no event could occur within 1 s of the previously detected event. For an ROI to be included, a noise threshold was empirically determined to avoid excessive false event detection: the standard deviation of the $\Delta F/F_0$ signal could not equal or exceed 0.0575. Additionally, ROIs were excluded if fluorescence was saturating, if they had drifted from the scan path such that signal was no longer detected, if they did not respond at least once to a supra-threshold stimulus (1.5 mA) delivered 10 times at the end of the experiment, and if the ratio of non-evoked to evoked events detected at this suprathreshold stimulation intensity was greater than 4.5. These parameters were tested against multiple data sets from simultaneous calcium imaging and cell-attached

**Table 1.** Details of sample sizes.

Table showing source of sample sizes for each subfigure in the study. Ex: Fig. 2C shows N = 13 cells from 11 slices and 6 mice.

| Figure | Cells | Slices | Mice |
|---|---|---|---|
| 1D | 139 | 5 | 2 |
| 1E | 139 | 5 | 2 |
| 1F | 52 independent pairs | 5 | 2 |
| 2B | 6 | 5 | 3 |
| 2C | 13 | 11 | 6 |
| 3A | 22 | 21 | 12 |
| 3B | 23 | 21 | 12 |
| 3C | 24 | 20 | 12 |
| 3D | 24 | 20 | 12 |
| 3E | 24 | 20 | 12 |
| 3F | 24 | 23 | 12 |
| 4C | N/A | N/A | 21 |
| 4E | N/A | N/A | 21 |
| 5A | N/A | N/A | 11 |
| 5C | 66 | N/A | 11 |
| 5E | 23 | N/A | 11 |
| 5F | 43 | N/A | 11 |
| 5G (CNO) | 23 | N/A | 5 |
| 5G (Vehicle) | 20 | N/A | 6 |
| 1 - figure supplement 1A | 6 | 6 | 1 |
| 1 - figure supplement 1B | 139 | 5 | 2 |
| 1 - figure supplement 2 (IEM) | 8 | 8 | 6 |
| 1 - figure supplement 2 (Veh) | 8 | 8 | 5 |
| 2 - figure supplement 1B | 4 | 4 | 2 |
| 2 - figure supplement 1C | 6 | 6 | 4 |
| 2 - figure supplement 1E | 5 | 5 | 2 |
| 2 - figure supplement 1F | 8 | 7 | 1 |
| 3 - figure supplement 1A | N/A | N/A | 16 |
| 3 - figure supplement 1B | N/A | N/A | 16 |
| 3 - figure supplement 1C | N/A | N/A | 12 |
| 3 - figure supplement 1D | N/A | N/A | 12 |
| 3 - figure supplement 1E | 24 | 20 | 12 |
| 3 - figure supplement 1F | 24 | 20 | 12 |
| 3 - figure supplement 1G | 24 | 20 | 12 |
| 3 - figure supplement 1H | 24 | 20 | 12 |
| 3 - figure supplement 1I | 21 | 19 | 12 |
| 4 - figure supplement 1 | N/A | N/A | 21 |
| 5 - figure supplement 2A (Pre CNO) | 23 | N/A | 5 |
| 5 - figure supplement 2A (Pre Vehicle) | 20 | N/A | 6 |
| 5 - figure supplement 2B (Pre CNO) | 23 + 23 rate-matched simulations | N/A | 5 |
| 5 - figure supplement 2B (Pre Vehicle) | 20 + 20 rate-matched simulations | N/A | 6 |

DOI: https://doi.org/10.7554/eLife.26231.016

recording experiments as previously reported (*O'Hare et al., 2016*) and were found to create an optimal balance of minimizing false detections and maximizing correct detections. Finally, slices which displayed poor loading, likely due to poor slice health or experimenter error during bulk-loading, were excluded from analysis. Each slice was required to have at least 12 SPNs of each subtype that passed all other exclusion criteria. This criterion was determined by finding the *N* at which coefficient of variation (CV) became a linear function of sample size, i.e. decreased only due to the CV denominator and not undersampling.

To analyze a pre-post effect within cell, such as wash-in of IEM-1460, only ROIs which were present and passed exclusion criteria both in pre and post recordings were included in analysis (See *Figure 1C* for matching ROIs before and after). Thus, drug effect was calculated for each individual cell using an internal baseline. Spike probability was calculated as the fraction of trials in which an evoked response was detected. Amplitude of an evoked calcium transient was calculated as the maximum $\Delta F/F_0$ in the transient waveform. Latency was calculated as the time between stimulus delivery and the time at which the cross-correlation dot-product reached half that of the perfect fit, i.e. the time of peak detection. When calculating dSPN/iSPN ratios, SEM was derived as: $\log_2\left(\frac{dSPN}{iSPN}\right)\sqrt{\frac{CV^2_{dSPN}+CV^2_{iSPN}}{N}}$. All analysis functions were custom-made in MATLAB unless otherwise noted.

To classify SPNs as 'high-firing' or 'low-firing' prior to application of IEM-1460 (*Figure 1D*), baseline calcium transient amplitudes for each SPN subtype were separated into two clusters according to a Gaussian mixture model (GMM). The effect of IEM-1460 was then calculated separately for 'high-firing' and 'low-firing' SPNs of each subtype and significance was determined using paired t-tests. In fitting the GMMs for dSPNs and iSPNs, the only user-specified input was the number of clusters (k = 2).

## Statistics

F statistics were calculated using repeated measures analysis of variance. For within-cell comparisons, t statistics were calculated by paired, two-sided t-tests. Otherwise, unpaired, two-sided t-tests were used. For non-normal data sets, Mann-Whitney U tests were used. All r values were obtained using Pearson correlation analyses. Normality was measured using the Kolmogorov-Smirnoff test of the data against a hypothetical normal cumulative distribution function. Unless otherwise indicated (e.g. *Figure 2C*, right panel), N values denote number of replicates considered biologically distinct for statistical measures (*Blainey et al., 2014*) (see *Table 1* for further detail). Technical replicates within a single biological sample were averaged to obtain a single value. For all statistical tests, confidence interval was set to $\alpha = 0.05$.

## Acknowledgements

The authors thank L Glickfeld, M Rossi, and MB Branch for their productive discussions and comments on the manuscript. The authors thank S Sternson for providing the *CAG*-FLEX-*rev*-hM4D:2a: GFP plasmid and the UNC Viral Vector Core for production of viruses. We gratefully acknowledge the following sources of funding: NS064577 and ARRA supplement (NC), AA021075 (HY), DA040701 (HY), McKnight Endowment Fund for Neuroscience (NC, HY), GM008441-23 (JO), NS051156 (KA), The Brain and Behavior Foundation (KA), The Tourette Association of America (KA) and the Ruth K. Broad Foundation (JO).

## Additional information

### Funding

| Funder | Grant reference number | Author |
| --- | --- | --- |
| National Institute of Neurological Disorders and Stroke | NS064577 | Nicole Calakos |
| National Institutes of Health | ARRA supplement to NS064577 | Nicole Calakos |

| National Institute on Alcohol Abuse and Alcoholism | AA021075 | Henry Yin |
|---|---|---|
| National Institute on Drug Abuse | DA040701 | Henry Yin |
| McKnight Foundation | | Henry Yin Nicole Calakos |
| National Institute of Neurological Disorders and Stroke | NS051156 | Kristen Ade |
| National Institute of General Medical Sciences | GM008441 | Justin K O'Hare |
| Brain and Behavior Research Foundation | | Kristen Ade |
| Tourette Association of America | | Kristen Ade |
| Ruth K. Broad Biomedical Research Foundation | | Justin K O'Hare |

The funders had no role in study design, data collection and interpretation, or the decision to submit the work for publication.

## Author contributions

Justin K O'Hare, Conceptualization, Data curation, Software, Formal analysis, Investigation, Visualization, Methodology, Writing—original draft, Writing—review and editing; Haofang Li, Resources, Investigation, Methodology; Namsoo Kim, Data curation, Formal analysis, Investigation, Methodology; Erin Gaidis, Data curation, Investigation, Methodology; Kristen Ade, Software, Validation, Methodology; Jeff Beck, Formal analysis, Methodology; Henry Yin, Conceptualization, Resources, Supervision, Funding acquisition, Methodology, Writing—original draft, Writing—review and editing; Nicole Calakos, Conceptualization, Resources, Supervision, Funding acquisition, Methodology, Writing—original draft, Project administration, Writing—review and editing

## Author ORCIDs

Justin K O'Hare http://orcid.org/0000-0002-7363-6064
Nicole Calakos http://orcid.org/0000-0002-9918-3294

## Ethics

Animal experimentation: All experiments were carried out under approved animal protocols (A112-17-04 & A263-16-12) in accordance with Duke University Institutional Animal Care and Use Committee standards.

## Decision letter and Author response

Decision letter https://doi.org/10.7554/eLife.26231.018
Author response https://doi.org/10.7554/eLife.26231.019

# Additional files

## Supplementary files

• Transparent reporting form
DOI: https://doi.org/10.7554/eLife.26231.017

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
