## [Decision Letter]

[Editors’ note: this article was originally rejected after discussions between the reviewers, but the authors were invited to resubmit after an appeal against the decision.]

Thank you for submitting your work entitled "Striatal fast-spiking interneurons drive habitual behavior" for consideration by *eLife*. Your article has been reviewed by two peer reviewers, and the evaluation has been overseen by a Reviewing Editor and a Senior Editor.

Our decision has been reached after consultation between the reviewers. Based on these discussions and the individual reviews below, we regret to inform you that your work will not be considered further for publication in *eLife*. All involved expressed great interest in your work, but reviewers noted some serious limitations that would require new experiments and/or extensive reanalysis (which were amplified during the consultation session amongst reviewers). We believe would likely require more work than is typical for an invited resubmission to *eLife*.

*Reviewer #1:*

This manuscript aims to uncover the role of striatal FSIs in habit formation. An extensive set of findings from different experimental approaches is presented, including (1) *ex vivo* slice work using cell-attached recordings and 2-photon imaging to examine the physiological changes in FSIs in habitual vs. non-habitual animals, (2) a behavioral study examining the effects of FSI manipulations on behavior, and (3) *in vivo* recordings examining the effects of FSI manipulation on striatal activity. Combining the results from these different experiments, the authors suggest that increased FSI activity accompanies habit formation, which in turn enables a specific set of SPNs to become active.

An obvious strength of the work is the combination of techniques to provide a truly multimodal perspective on FSI changes. This integrative approach to FSI function is novel and innovative: the in vitro and modeling literature has identified FSIs as powerfully shaping striatal activity, yet not much is known about their functional role *in vivo* and habitual behavior in general. Thus, this work has the potential to provide a major step forward in uncovering the neural substrates of habits, an issue of widespread interest.

Unfortunately, I found serious issues with most of the individual experiments in the manuscript, most obviously with the *in vivo* recordings and analysis (with which I am most familiar) but also in the other components. I am supportive of the authors' integrative approach in principle, and the conclusions advanced in the manuscript may well turn out to be correct; however, as things stand, there are too many missing controls, confounded analyses, and alternative interpretations for me to have confidence in the conclusions.

Specifically:

1) I am most qualified to comment on the *in vivo* recording studies. There are some major flaws in the experimental design (missing controls) and analysis, and as a result there are several possible alternative interpretations. In particular:

1a) It appears the *in vivo* recording comparison between vehicle and CNO was always run in the same order (vehicle followed by CNO). This introduces an obvious confound that time, or variables correlated with it, mediate some of the observed differences. The authors should control for this by counterbalancing the order across days, doing only one session per day and alternating between CNO and vehicle, or having a washout comparison following the CNO. A less convincing approach that would at least demonstrate awareness of this issue is to fit statistical models that include time as a regressor, using multiple regression (e.g. for Figure 1 and Figure 5) – do the drug effects still hold?

1b) There is no description of what the animals are doing during the recordings. Striatal firing patterns are known to be correlated with behaviors such as grooming, running, and sleep (Berke et al., Neuron 2004; Burguiere et al. Curr Op Neuro 2015). The authors need to make sure that their reported effects are not due to differences in the expression of these behaviors. Were video data collected to rule this out?

1c) The identification of rhythmic signatures in those SPNs that appear to be most affected by FSI changes is potentially exciting. However, the analysis used (proportion of ISIs) to identify rhythmic activity is flawed and should be replaced by something more appropriate (see below). What the authors are doing is confounded by firing rate: more active neurons are more likely to have ISIs in higher frequency ranges. Thus, the result reported that higher firing rate neurons have more gamma ISIs is expected based on statistical properties alone. To conclude that this reflects a physiological property of interest, the authors need to use either spike-field measures of phase locking (e.g. Bokil et al. J Neurosci 2010) or compare the observed ISI distribution in specific bands to that obtained from a firing rate matched Poisson process (see e.g. Kass et al. Analysis of Neural data book).

2a) The behavioral part of the paper, on which FSI inhibition with DREADDs results in less habitual behavior, is novel but invites a simpler explanation than that provided by the authors: any disruption of DLS activity would interfere with habitual behavior. This would be consistent with the substantial literature on this from lesion and local infusions (e.g. Packard & McGaugh 1996; Yin et al. 2004).

2b) Comparing the RRshort and RI groups, it seems there is a difference in total number of lever presses, and potentially in the number of rewards earned (Figure 3—figure supplement 1). The authors should examine if differences in these behavioral measures are correlated with their physiological variables; if they do, that would call into question the interpretation that habit is the key factor here.

3) I am not an expert on the slice and 2P components of the paper, but here too I have some concerns about controls and interpretation:

3a) I found it notable that there was no control for time (through counterbalancing or washout). As a result, how can the authors be confident that results like Figure 1 do not simply reflect nonspecific changes over time such as homeostasis?

3b) If I am understanding correctly how iSPNs were identified – through the absence of DrD1a-tdTomato – this means interneurons including FSIs are included in this category. If this is correct, I don't see how the shift in direct/indirect pathway latency, and other results that claim a difference between dSPN and iSPN groups, are supported by the data. I realize this method is commonplace when you want to make a statement about dSPNs vs iMSNs and have no reason to think anything is changing in FSIs, CINs etc. But in this study we are provided explicit evidence that there are systematic changes happening in FSIs, so there is going to be a clear bias in a measurement that has those in the pool.

3c) In the IEM-1460 part of the paper, the authors could better motivate their logic about selectivity to FSIs by documenting other work that AMPA receptors in striatal FSIs lack GLuA2 subunits, while SPNs AMPARs have GluA2, and that IEM-1460 does not affect glu signaling in MSNs (most clearly demonstrated by Gittis et al. 2011 JNsci).

Otherwise their conclusion would seem to require careful voltage-clamp experiments showing the effects of this agent on identified AMPA currents in FSIs and SPNs.

3d) In the PV-Arch experiments and Figure 2—figure supplement 1 in particular: given that there is an effect of laser on SPN firing, again, how can the authors claim that FSI changes are what underlies their results? I now understand that this is in fact consistent with the mechanism the authors are proposing, but was confused by their use of the term "off-target" which to me seemed to imply that they wanted to verify that their stim affected only FSIs and not SPNs directly – which would be useful to test under synaptic blockage conditions, but that isn't what they actually do. I would find an expanded description in the text helpful.

*Reviewer #2:*

The study by O'Hare et al. provides timely evidence for how PV+ FSI in the dorsal striatum influence the activity of direct and indirect SPNs. The authors nicely demonstrate that the key circuit modifications performed by FSI map onto previously described activity patters for habitual-vs-goal directed behaviors. They argue that FSIs undergo plastic changes during habit formation, and influence a subset of SPNs to promote the expression of habitual behavior. I am generally enthusiastic about this study. I thought the manuscript was presented with a logical flow, and easy to understand. The experimental methods and analyses used are OK, and I do not foresee the necessity of additional experimental work.

General comments:

The authors provide evidence for acute functional reorganization of the microcircuit upon application of IEM-1460. Specifically, a change in dSPN and iSPN response magnitude and latency (Figure 1). It will be useful to report the actual (ms) change in latency (instead of a log-relative measure) because it will facilitate comparison with other work (including O'Hare et al. Neuron 2016), and to assess if these changes are a result of a delay in dSPN responding or speeding of iSPN response.

Secondly, it appears that most of the arguments the authors make are essentially boil down to FSIs affecting the activity of a subset of 'high firing' or very active d/ and iSPNs. Despite this recurring observation (in pharmacological, optogenetic, ex-vivo and *in vivo* experiments), the authors do not speculate about the significance of this particular aspect of their results. Concretely, naive slices contain SPN responses that have a distribution of amplitudes (1D) and slices from habitual animals have a large-amplitude, indirect-SPN-delay biased responses. How, exactly, do the authors envision FSI recruitment (or lack thereof) during habit formation or goal directed behaviors shape the naive SPN responses into what is cartoonishly depicted in Figure 1. In other words, I am asking what the authors' framework is for striatal ensamble activity that underlies habit beyond the clear cell-by-cell results they provide.

It is odd that the authors focus their physiology experiment (Figure 3 onwards) to mice that behaved in stereotyped 'habit' or 'goal-directed' ways (shaded blocks in Figure 3—figure supplement 1). I did not notice a justification for this in the text. Is it the case that goal-directed mice in RI group (excluded dots at bottom) exhibited physiological characteristics that were similar to RR group, or alternatively naive mice? How are the cellular physiology results changed if those animals were included? In a previous publication from the group, they combined all mice, and formed a continuous distribution of goal-vs-habit mice.

[Editors’ note: what now follows is the decision letter after the authors submitted for further consideration.]

Thank you for resubmitting your work entitled "Striatal fast-spiking interneurons drive habitual behavior" for further consideration at *eLife*. Your revised article has been favorably evaluated by a Senior editor, a Reviewing editor, and two reviewers.

The manuscript has been improved but there are some remaining issues that need to be addressed before acceptance, as outlined below:

1) In this revision the authors have made several essential clarifications and performed additional control experiments and additional analyses, which have strengthened the study in important respects. Despite these improvements, several experiments in my mind are still missing the correct analyses or controls; details follow below. As before, I am enthusiastic about the integrative approach taken by the authors in combining and synthesizing across multiple experimental approaches. However, I detect an overall tendency to constrain the interpretation of specific experiments by referring to an overall synthesis or working model derived from multiple experiments. As I elaborate on below, for me to have more confidence in the results, I expect the authors to first carefully interpret the results from specific individual experiments in the Results section, without reference to an overall favored working model. Then, they are encouraged (in the Discussion) to come up with a synthesis across experiments. What I think they should avoid is writing parts of the results as if this synthesis has already been shown to be correct. This overall issue is illustrated below, particularly in my response to points 3a and 3b.

1a) For the order of vehicle and CNO applications, I assumed the worst because there was no mention of session ordering in the original manuscript, and the Figure 1 experiments employed a fixed control followed by treatment order (see my comments on this below). The fact that vehicle/CNO treatments were given on separate days and counterbalanced for order is important information that should be included in the Materials and methods, ideally with the table shown in the rebuttal, so that readers can see that in some cases, there were 10+ weeks between the two experiment days.

1b) The new inclusion of locomotion data, which show similar distance traveled for both groups, is useful and increases confidence in the interpretation of the DREADD experiments.

1c) I apologize for any confusion that my original description may have caused. My point here is quite simple: what is the evidence that relative gamma ISI density, rather than firing rate, is the relevant variable here? In the rebuttal, the authors provide a figure showing a lack of significant correlation between firing rate and the fraction of gamma-band ISIs. But this does not exclude the possibility that if the new Figure 5 would test for a correlation between firing rate (fold change; the quantity plotted on the vertical axis) and baseline firing rate instead of gamma density, the reported correlation would persist. I agree with the points made by the authors that it's entirely possible that low overall firing rates are responsible for the lack of a significant correlation between firing rate and the fraction of gamma-band ISIs, and the Poisson analysis is interesting in its own right (my original suggestion would require a further step: asking if the excess gamma ISI probability predicts firing rate change following CNO). However, these are all indirect points. The most direct way of testing whether invoking gamma ISI density is necessary is to determine if using baseline firing rate yields the same overall result or not. If so, I don't see the need for a more complicated, gamma-ISI related explanation. In the rebuttal, the authors state, "fold change correlated with gamma density, not the rate itself" but I did not see evidence for the latter claim in the manuscript: this needs to be tested and stated explicitly.

2a) Acknowledging the interpretation that the FSI manipulations performed are in effect a disruption of DLS activity, as the authors do now in the Discussion, is appropriate. However, I believe the authors' suggestion that their work is the first to show an effect of DLS manipulation on the expression (rather than acquisition) of habits is overstated. Packard and McGaugh (1996) performed lidocaine injections in the caudate nucleus following acquisition of a response strategy on a plus maze. Post-training baclofen/muscimol infusions into the DLS reduce habitual lever-pressing for alcohol (2012 Biol Psychiatry, 2016 Frontiers Behav Neuro), and has similar effects on lever pressing for cocaine (Zapata et al. J Neurosci 2010). Thus, for different operationalizations of habit and reinforcer, post-training manipulations of DLS have been shown to affect habitual (but not goal-directed) behavior.

2b) Great, inclusion of the lever press and total rewards data strengthens the results.

3a) I appreciate the authors are combining multiple approaches and lines of evidence to converge on a proposed mechanism – this is a strength of the paper. But in this case it's hard to see beyond the most direct way to rule out nonspecific changes related to time, which is simply to counterbalance the order of the treatment and control groups, or to collect washout data. The authors second observation provided is the most relevant in that it does contain a treatment vs. control comparison without the potential time confound, but this is a separate experiment, but low n, where we are not told how its timescale maps on to the main experiment, while leaving open whether the effect shown here underlies the full Figure 1 results. The other observations are more indirect still. Fixing this would require re-doing the experiment. However, if the authors feel strongly that they can argue against this confound, this flaw may be outweighed by intriguing results in other parts of the paper that collectively contribute to an emerging new view of FSI-SPN interactions.

3b) As with 3a, I note that the authors' style is to make inferences from multiple experiments to converge on an overall explanation. This is admirable and important. However, this does not absolve them of responsibility in accurately describing what can and cannot be concluded from each experiment separately. I do not think it is appropriate to take their proposed overall mechanism and use it as a source of top-down constraints on the initial interpretation of specific, individual experiments. What I want to see is careful descriptions and conclusions for each individual experiment in the Results that appropriately take the limitations of each into account. Then, the authors are encouraged to integrate the results into an overall proposed mechanism, and if they wish, use the resulting synthesis to inform interpretation of specific pieces of data – but not where those experiments and data are first described.

For this particular point, while it may be true that about 1% of striatal neurons are FSIs, they tend to be highly active, rendering them more likely to be included in recording studies (and, depending on how cells are identified in imaging studies, in those too; see e.g. Harris et al. Nat Nat Neurosci 2016). I agree with the authors that results such as Figure 1 are unlikely to result from inadvertent inclusion of a few FSIs because we are shown the full distribution of data points. However for others such as Figure 1 which simply compare means, it seems possible that a few potential FSI outliers could shift the means significantly.

3c) This additional explanation is helpful.

3d) Great, this new figure fully addresses my concern.

4) The comments related to behavior-physiology studies (Figure 3) were circumnavigated. The authors DO report an exclusion criterion, namely; "RRshort-trained mice with NDLPr <0 were considered to be goal-directed…Mice not meeting inclusion criterion were not used…". But, as part of their conclusions they routinely refer to the task as " habit promoting reinforcement protocol", and "habit-trained subjects…". Figure 3—figure supplement 1 indicates that 4 of 9 mice trained in this task do not engage in a habitual behavior, an insignificant 44% of mice.

In Figure 4, the authors exposed mice to the behavioral protocol that already produces ~44% of goal directed behavior, to argue that DREADD manipulation produces robust goal directed behavior.

So, without a devaluation test before CNO day to assess degree of habit induced, it is hard to accept the claim that CNO promoted less habitual behavior. If the physiological phenotype of the intermediate animals (trained on task, but did not exhibit habitual behavior) were known, I would be more convinced that CNO inactivation of FSI would (at least) shape an incomplete phenotype described in Figure 3 really think Figure 4 has compelling results, and hope the authors address this concern.

But as it stands now, I am not fully convinced that the hm4D manipulation produces less habitual behavior. The author could come up with a clever, but robust statistical approach to assess/argue more convincing statistical result. For example if they serially excluded 40% of subjects in both groups, and performed the test, what fraction of the comparisons are significant? Or another formulation of shuffle tests. As I pointed out, a within-subject experiment would have been most ideal; that is each animal (in both PV-hM4D and OV-GFP groups) getting two devaluation sessions with and without CNO. If the authors cannot address this weakness via reanalysis of existing data, I think they will have to demonstrate a more convincing behavioral result with a new cohort of mice.

[Editors' note: further revisions were requested prior to acceptance, as described below.]

Thank you for resubmitting your work entitled "Striatal fast-spiking interneurons drive habitual behavior" for further consideration at *eLife*. Your revised article has been favorably evaluated by a Senior editor, a Reviewing editor (Michael Frank), and one reviewer.

We have sent the manuscript back to the one reviewer who had more reservations (the other reviewer was largely satisfied in the last round and we determined that you have addressed their remaining issues). The manuscript has been improved but there are now just a few remaining issues that need to be addressed before acceptance, as outlined by the reviewer below:

This resubmission contains several additional control experiments and analyses, which collectively address three major issues identified as remaining after the last round of reviews, with mixed success. One case is convincing, the second exacerbates my concerns about the authors' interpretation, and the third appears reasonable but is not incorporated into the manuscript.

– My request to control for the effects of time in the IEM-4160 slice experiments (Figure 1) is fully satisfied by the addition of new, across-group experiments for slices pre-incubated with IEM and vehicle.

– The authors claim that gamma-band ISIs are predictive of firing rate changes in SPN firing following CNO. However, pre-CNO firing rate is a numerically better predictor of such changes than excess gamma-band ISIs (r = -0.61 vs -0.45), casting doubt on the relevance of this feature given that a simpler and more predictive measure is available. The authors argue that firing rate "non-specifically" predicts such changes (i.e. for both vehicle and CNO) and that therefore excess gamma-band ISIs (which only predict post-CNO firing rates, but not post-vehicle rates) is an interesting quantity; however, these results could also be interpreted to mean that gamma-band ISIs is simply a worse predictor overall. I could imagine additional analyses, such as a multiple regression, that could help clarify this issue; however, regardless of the specific outcome, any remaining variance explained will likely be low and of unclear relevance. In addition, the authors should be aware that a comparison between two t-tests (one significant, one non-significant) does not constitute sufficient evidence for an interaction (see Nieuwenhuis et al. Nat Neurosci 2011) and report the correct test instead.

– In order to address the concern that FSIs included in the iSPN dataset (which is defined by the absence of D1R label) may contribute to the effects in Figure 1, the authors provide new data suggesting this is unlikely. At a minimum the authors need to discuss this possible confound in the manuscript, and ideally include the data shown in a supplementary Figure.

---

## [Author Response]

[Editors’ note: the author responses to the first round of peer review follow.]

Reviewer #1:[…] 1) I am most qualified to comment on the in vivo recording studies. There are some major flaws in the experimental design (missing controls) and analysis, and as a result there are several possible alternative interpretations. In particular:1a) It appears the in vivo recording comparison between vehicle and CNO was always run in the same order (vehicle followed by CNO). This introduces an obvious confound that time, or variables correlated with it, mediate some of the observed differences. The authors should control for this by counterbalancing the order across days, doing only one session per day and alternating between CNO and vehicle, or having a washout comparison following the CNO. A less convincing approach that would at least demonstrate awareness of this issue is to fit statistical models that include time as a regressor, using multiple regression (e.g. for Figure 1 and Figure 5) – do the drug effects still hold?

When subjects received both vehicle and CNO injections, the order of these sessions was counterbalanced and the tests occurred on separate days. Below is a summary table for reviewers’ convenience. The authors were unable to locate the source of confusion in the manuscript – please advise if there is language explicitly indicating that Vehicle was always administered before CNO.

**Mouse ID****CNO****Vehicle**9695/4/169718/26/168/24/169728/24/1615215/4/167/25/1615225/4/167/25/1615235/4/1685118/24/1685128/24/1685138/26/168/24/16

1b) There is no description of what the animals are doing during the recordings. Striatal firing patterns are known to be correlated with behaviors such as grooming, running, and sleep (Berke et al., Neuron 2004; Burguiere et al. Curr Op Neuro 2015). The authors need to make sure that their reported effects are not due to differences in the expression of these behaviors. Were video data collected to rule this out?

Yes, 3D motion tracking data were recorded for all mice during these open field recordings and we now include these data as part of Figure 5. We observed no effect of treatment (CNO versus vehicle in PV‐hM4D mice) on locomotion.

Given similarities in overall locomotion, it is unlikely that systematic biases in sleep or grooming could account for group‐wise differences. Our lab routinely studies the overgrooming “OCD” model referenced above. We regularly observe an inverse relationship between grooming and distance travelled (as in Ade et al., 2016) because mice stop walking when they groom, as well as sleep.

1c) The identification of rhythmic signatures in those SPNs that appear to be most affected by FSI changes is potentially exciting. However, the analysis used (proportion of ISIs) to identify rhythmic activity is flawed and should be replaced by something more appropriate (see below). What the authors are doing is confounded by firing rate: more active neurons are more likely to have ISIs in higher frequency ranges. Thus, the result reported that higher firing rate neurons have more gamma ISIs is expected based on statistical properties alone. To conclude that this reflects a physiological property of interest, the authors need to use either spike-field measures of phase locking (e.g. Bokil et al. J Neurosci 2010) or compare the observed ISI distribution in specific bands to that obtained from a firing rate matched Poisson process (see e.g. Kass et al. Analysis of Neural data book).

We thank Reviewer #1 for their diligence in considering sources of potential caveats in analysis and interpretations – we are always appreciative of opportunities to avoid erroneous conclusions. On this particular point though, there appears to be some error in interpreting the manuscript.

Reviewer #1 re‐states our result as “Thus, the result reported that higher firing rate neurons have more gamma ISIs” but this is not our observation. Instead, we only found that the degree of modulation of firing, ie fold change, correlated with gamma density, not the rate itself. However, the reviewer’s interpretation has helped us recognize a better way of labeling the y‐axis. We have removed the phrase “firing rate (fold change)” and replaced it with “rate modulation (fold change)”, as the former may have been misleading. We have also relabeled the x‐axis from “Baseline relative gamma density” to “Fraction gamma ISIs, baseline” because a higher number does not mean “more gamma ISIs” in the absolute sense. Other than these changes to graph labels, statements in the legend and main manuscript were accurate.

We have also added two further analyses for the reviewer.

1) We find that there is no significant association between firing rate and fraction of gamma range ISIs in our data.

2) We perform the suggested Poisson process comparison, and find that the result continues to support the significance of gamma frequency ISIs as a distinct SPN activity feature above and beyond what would occur by chance due to overall firing rate.

Author response image 1 shows new analysis of firing rate (*not rate modulation*) vs. gamma ISI density. No association is noted (r = 0.25, p = 0.11), likely because the mean frequencies in our dataset are so low to begin with (0.01‐2.5 Hz) that the sort of temporal crowding that would drive such a relationship is not significantly present.

Author response image 2 compares Poisson process‐simulated distribution of ISIs (blue) compared to actual observations (red). As we now describe in Materials and methods, each individual SPN was simulated by a Poisson process matched to its own firing rate. SPNs consistently displayed more gamma‐frequency ISIs than expected by a Poisson point process (for both CNO‐ and vehicle‐treated cohorts).

**Author response image 2. respfig2:** 

2a) The behavioral part of the paper, on which FSI inhibition with DREADDs results in less habitual behavior, is novel but invites a simpler explanation than that provided by the authors: any disruption of DLS activity would interfere with habitual behavior. This would be consistent with the substantial literature on this from lesion and local infusions (e.g. Packard & McGaugh 1996; Yin et al. 2004).

We thank the reviewer for highlighting the alternative interpretation that non‐specifically disrupting DLS activity could affect habit expression and we now acknowledge this point in the main text Discussion section.

We do not think this observation lessens the impact of our findings for three major reasons.

1) This is a frequent unavoidable caveat for a large number of studies of this nature. Following this 3 point summary, we describe specific examples from two recent, high impact studies to make this point concretely.

2) More importantly, the support for FSIs as a mechanism for habit expression in this study does not rest solely on this in vivoperturbation of behavior experiment, but rather the larger body of findings throughout the manuscript prior to this key test of the hypothesis. In this regard, our findings are stronger than several other high impact studies that only show in vivobehavioral perturbation as evidence for a specific role. Briefly, in addition to eliminating habit by reducing FSI activity, we find that: (1) Pharmacological and optogenetic reduction of FSI activity modulates ex vivostriatal SPN firing properties that correlate with habitual behavior (Figure 1, Figure 1—figure supplement 2, and Figure 2) (correlations originally reported in O’Hare et al., 2016). (2) Long‐lasting increases in FSI excitability are found as a consequence of habit learning in mice, in comparison to goal‐directed mice (Figure 3). In the revised Discussion section, we discuss the supporting nature of these data. We think the revised Discussion benefits from discussing this important point, and thank the reviewer for raising it.

3) An aspect of the novelty of our findings was overlooked. To date, lesions of DLS and their impact on habit have only studied lesion effects on *learning/acquiring* the behavior. This study, therefore, contributes a first demonstration that a post‐learning DLS manipulation can prevent the *expression* of an acquired habit.

Copies of key data from manuscript referenced in Point 2 above appear in Figure 1, Figure 2 and Figure 1—figure supplement 1:

Specific discussion of two published comparable studies:

a) Witten, et al. (2010, Science) found that silencing cholinergic interneurons (CINs) of the nucleus accumbens (NAc) disrupts cocaine conditioning in mice. However, an alternative interpretation exists that a general disruption of NAc activity would yield a similar result. Indeed, in the second‐to‐last paragraph of the main text, the authors note that their result is more similar to that seen from general, less‐specific NAc manipulations relative to studies which chronically ablated CINs:

“Together, these data demonstrate that selectively inhibiting ChAT interneurons in the NAc with high temporal precision has the overall effect of increasing MSN activity and blocking cocaine conditioning in freely moving mammals. These behavioral results do not support conclusions arising from chronic ablation of the cholinergic interneurons (20); instead they are more consistent with interpretations arising from faster but less cellularly targeted pharmacological modulation in the NAc”.

In addition, as mentioned in point (3), unlike the above study’s context in cocaine conditioning, our result of a role of DLS in habit expression has not been reported in the literature using less‐specific manipulations, to our knowledge.

b) A recent study also examining the role of PV+ interneurons in dorsolateral striatum by Lee, et al. (2017, Neuron) uses opto‐ and chemo‐genetic inhibition of FSI activity to show a role for these cells in early‐phase Pavlovian conditioning. For example, “F” in Author response image 3 shows the study’s positive result of a change in hit rate with chemogenetic inhibitory DREADD expression in DLS FSIs (light blue).

**Author response image 3. respfig3:** 

A legitimate claim could be levied on this study that this result in hit rate might also occur with a less‐specific DLS manipulation. No general manipulation of DLS activity was made to assess this possibility.

The authors do provide some demonstration of behavioral specificity by showing that their manipulation has no effect on false alarm rate (licking behavior to a non‐conditioned stimulus) in subfigure G of Author response image 3. We essentially perform the analogous control in our task, showing that chemogenetically inhibiting PV cells in DLS does not alter general lever pressing behavior by evaluating behavior in the non‐devalued probe test (Figure 4—figure supplement 1).

2b) Comparing the RRshort and RI groups, it seems there is a difference in total number of lever presses, and potentially in the number of rewards earned (Figure 3—figure supplement 1). The authors should examine if differences in these behavioral measures are correlated with their physiological variables; if they do, that would call into question the interpretation that habit is the key factor here.

The reviewer raises an important concern. A key feature of our chosen training protocol is that it allows us to make training experiences as similar as is practically possible apart from the feature of habitualness. We present the requested data in the revised manuscript (Figure 3—figure supplement 1). We also refer the reviewer to our prior analysis of this question in a separate cohort in which we also found that lever press rate was insufficient as a variable to predict striatal physiological properties (O’Hare et al., Neuron, 2016).

As a side point, we also asked whether the bimodal distribution of rewards for the goal-directed subjects corresponded to the bimodally‐distributed response durations displayed by FSIs from goal‐directed mice in response to somatic current injections shown in Figure 3. However, we found this not to be the case. In fact, FSIs falling into both modes were found within single goal‐directed subjects. For example, two FSIs were recorded from one goal-directed mouse with response durations of 494.7 ms and 180.9 ms.

3) I am not an expert on the slice and 2P components of the paper, but here too I have some concerns about controls and interpretation:3a) I found it notable that there was no control for time (through counterbalancing or washout). As a result, how can the authors be confident that results like Figure 1 do not simply reflect nonspecific changes over time such as homeostasis?

We agree with this concern and would like to clarify how we approached obtaining additional evidence to assure ourselves that the result was not spurious due to a technical artifact related to time that passed between pre and post measures. Instead of a vehicle control to account for the variable of time, we chose to perform two additional ephys experiments because this choice offers the advantage of making the same observations multiple times using orthogonal techniques. When we found an effect of IEM drug, instead of repeating the experiment with vehicle, we opted to pursue more direct electrophysiological measures (Figure 1—figure supplement 2, Figure 2) and more specific manipulations (photoinhibition, Figure 2) to prospectively test the Figure 1 result.

We believe that these three results combined remove doubt for the potential of the original observation to be an artifact – all three experiments show that FSIs increase evoked SPN responses, and specifically multi‐AP responses, but not unitary.

We also note that if the above Arch data comparing laser on vs off are presented in a manner analogous to the Figure 1 2PLSM data (laser ON – OFF) shown earlier, the result is strikingly similar – analysis shown in Author response image 4 for reviewer’s benefit:

**Author response image 4. respfig4:** 

3b) If I am understanding correctly how iSPNs were identified – through the absence of DrD1a-tdTomato – this means interneurons including FSIs are included in this category. If this is correct, I don't see how the shift in direct/indirect pathway latency, and other results that claim a difference between dSPN and iSPN groups, are supported by the data. I realize this method is commonplace when you want to make a statement about dSPNs vs iMSNs and have no reason to think anything is changing in FSIs, CINs etc. But in this study we are provided explicit evidence that there are systematic changes happening in FSIs, so there is going to be a clear bias in a measurement that has those in the pool.

The reviewer again makes an astute point. We had considered this possibility and concluded that it was not a tenable explanation. We relay our thinking here. First, CINs and some FSIs get excluded *a priori* from analysis based on size exclusion criteria we set up in developing the imaging approach initially (O’Hare et al., 2016). In that development phase, we used reporter cell lines to define properties of non‐SPNs and developed exclusion criteria to limit their contamination of the data set. This size criterion is noted in the current manuscript’s Materials and methods section. Second, even without any exclusion criteria, FSIs comprise only 1% of the striatal neurons, therefore, even in the most optimistic setting, the data would have to derive from at most 2‐3 cells among the 52 imaged cells that generated the observation. Such a minor fraction of cells is insufficient to generate the correlations we find (see for example Figure 1). Moreover, if the affected cells were FSIs, they should partition unequally between SPN types, predominantly in the “nonDrd1a” iSPN group (see Ade et al., Frontiers Neurosci. 2011), and this is also not the case (e.g. Figure 1). Third, subsequent electrophysiological experiments in SPNs confirmed by their electrophysiological signatures also show the finding, and in both SPN subgroups (Figure 1—figure supplement 2).

3c) In the IEM-1460 part of the paper, the authors could better motivate their logic about selectivity to FSIs by documenting other work that AMPA receptors in striatal FSIs lack GLuA2 subunits, while SPNs AMPARs have GluA2, and that IEM-1460 does not affect glu signaling in MSNs (most clearly demonstrated by Gittis et al. 2011 JNsci).Otherwise their conclusion would seem to require careful voltage-clamp experiments showing the effects of this agent on identified AMPA currents in FSIs and SPNs.

The paper that the reviewer alludes to (Gittis et al. 2011 JNsci) was referenced in the original manuscript. In the revision, we further expand this discussion to specifically mention GluA2 receptors.

Revised manuscript text: “To manipulate FSI activity, the calcium‐permeable AMPA receptor (CP‐AMPAR) antagonist IEM‐1460, which predominantly weakens excitatory synaptic inputs onto FSIs in striatum24, was used. Striatal FSIs express AMPARs lacking the GluA2 subunit, rendering them permeable to calcium 25, whereas SPNs do not typically express CP‐AMPARs. Consistent with this difference in AMPAR subunit expression, IEM‐1460 does not affect excitatory synaptic currents in SPNs but strongly decreases excitatory transmission onto FSIs24.”

3d) In the PV-Arch experiments and Figure 2—figure supplement 1 in particular: given that there is an effect of laser on SPN firing, again, how can the authors claim that FSI changes are what underlies their results? I now understand that this is in fact consistent with the mechanism the authors are proposing, but was confused by their use of the term "off-target" which to me seemed to imply that they wanted to verify that their stim affected only FSIs and not SPNs directly – which would be useful to test under synaptic blockage conditions, but that isn't what they actually do. I would find an expanded description in the text helpful.

The original manuscript did address this point (copied below). In addition, we have gone ahead and performed the straightforward suggested experiment under voltage‐clamp conditions. We again thank the reviewer for the overarching interest in ensuring our study is of the highest quality.

Original Legend of Figure 2—figure supplement 1:

”…(B) Left: recording configuration to assess off-target effects of 532 nm light on SPN firing. Middle and Right: SPN responses to somatic current injection with interposed 532 nm light as in (A). Although analysis of variance showed an effect of laser on SPN firing (F(1.04, 7.27) = 9.80, p = 0.015, n = 8), this effect was due to an early frequency adaptation which SPNs are known to display in response to suprathreshold excitation1. SPN firing rates during and after laser stimulation were indistinguishable (p = 0.31, n = 8).”

Revision, new data in Figure 2—figure supplement 1:

Under pharmacological block of NMDA, AMPA, and GABAA receptors, 532 nm light‐induced currents were measured in voltage‐clamp in SPNs and striatal FSIs of PV‐Arch mice. We observed large lightdriven currents in FSIs, but not SPNs.

Reviewer #2:General comments:The authors provide evidence for acute functional reorganization of the microcircuit upon application of IEM-1460. Specifically, a change in dSPN and iSPN response magnitude and latency (Figure 1). It will be useful to report the actual (ms) change in latency (instead of a log-relative measure) because it will facilitate comparison with other work (including O'Hare et al. Neuron 2016), and to assess if these changes are a result of a delay in dSPN responding or speeding of iSPN response.

We thank reviewer #2 for this suggestion. We now include mention of the absolute latency values in the revised. They are primarily informative for showing that both cell types have a trend toward altered latencies in which the effect is slightly greater in the dSPNs. However, we would also like to provide further explanation as to why we find the relative difference in latency more valuable than absolute values. Firstly, absolute values are influenced by electrode position in the brain slice relative to the imaged area, whereas the difference between two populations distributed about the same electrode bypasses that concern. Secondly, the absolute latency values include the superimposition of not only distance from electrode to soma, but also kinetics of calcium dye transients, and time differences related to the location of the cell along the line scan vector. Since all of these factors are consistent between the test and control conditions, significant differences in relative latency can be observed, it is just that the absolute values are of a bit less importance. Nonetheless, we are happy to provide values closer to the primary data and thank you for your interest.

In the present study, we find that IEM‐1460 did not significantly affect the latency of calcium transients in either SPN subtype; only the relative latency between these two pathways. We observed a non‐significant trend for both pathways to exhibit slower latencies with dSPNs showing a more pronounced delay. These data are now reported in the text and copied below. The finding of a significant effect for relative differences in latency, but not absolute latencies is also consistent with the results from our prior analysis in habit versus goal mice (O’Hare et al., 2016).

“IEM-1460 also changed the relative latency to fire between direct and indirect pathway SPNs by increasing the pre-existing bias in relative pathway timing whereby iSPNs tend to respond to cortical excitation more quickly than dSPNs (Figure 1) (mean absolute latency values for dSPNs: 144.03 ± 7.08 ms ACSF, 154.33 ± 7.92 ms IEM-1460, N = 87; iSPNs: 130.31 ± 7.87 ms ACSF, 134.43 ± 8.89 ms IEM- 1460, N = 52).”

Secondly, it appears that most of the arguments the authors make are essentially boil down to FSIs affecting the activity of a subset of 'high firing' or very active d/ and iSPNs. Despite this recurring observation (in pharmacological, optogenetic, ex-vivo and in vivo experiments), the authors do not speculate about the significance of this particular aspect of their results. Concretely, naive slices contain SPN responses that have a distribution of amplitudes (1D) and slices from habitual animals have a large-amplitude, indirect-SPN-delay biased responses. How, exactly, do the authors envision FSI recruitment (or lack thereof) during habit formation or goal directed behaviors shape the naive SPN responses into what is cartoonishly depicted in Figure 1. In other words, I am asking what the authors' framework is for striatal ensamble activity that underlies habit beyond the clear cell-by-cell results they provide.

We are pleased to learn of the interest in further speculation to address this important point and have added this to Discussion. We initially included speculative interpretation of a number of aspects of how the cellular findings would impact the broader circuitry. Those points are copied below as a reminder.

From the last paragraph of Results after describing our *in vivo* findings:

“While FSIs can have an overall strongly inhibitory effect *in vivo* on SPN firing as traditionally assumed, we also found evidence that they potentiate activity in a select population of high-gamma SPNs. This selective potentiation may be akin to a winner-take-all “focusing” mechanism that increases the signal-to-noise ratio in corticostriatal transmission. According to such a mechanism, the subset of recruited SPNs would be facilitated while the less-relevant, low-gamma SPNs would be suppressed.”

From Discussion paragraph 3:

“This *in vivo* finding is also consistent with a previous *in vivo* report that SPNs with weaker responses to cortical activity displayed the most marked disinhibition upon GABAA receptor blockade16. While directly comparing the subsets of excited SPNs identified in the slice and *in vivo* would not be technically straightforward, an important future direction will be to determine whether there are unique biological properties that set the excited subset of SPNs apart.”

From Discussion paragraph 5:

“Indeed, recent evidence suggests that spatially-clustered SPN activity encodes information relevant to locomotor behavior45. In habits, one possible mechanism then is that task-specific cortical commands drive46, or at least initiate47, high-frequency firing in a cluster/subset of SPNs that would then be preferentially excited by FSIs. Additionally, in such a mechanism, feed-forward inhibition of less-active SPNs16 by FSIs might then serve as a selective filter to further enhance signal-to-noise ratio in corticostriatal transmission. One testable prediction of this model is that different behaviors would reveal different subsets of high-gamma SPNs that are excited by FSIs.”

Additional discussion of model for how FSIs drive the phenomena depicted in Figure 1:

“Unexpectedly, we found that the directionality by which FSIs modulated these properties was opposite to our original hypothesis: instead of the expected disinhibition of SPNs, silencing FSIs reduced SPN output (Figure 1‐E). […] In this setting, incoming cortical activity would be predicted to recruit more FSI activity that would in turn drive more firing of SPNs and shift their latencies such that direct pathway SPNs would tend to fire relatively sooner.”

It is odd that the authors focus their physiology experiment (Figure 3 onwards) to mice that behaved in stereotyped 'habit' or 'goal-directed' ways (shaded blocks in Figure 3—figure supplement 1). I did not notice a justification for this in the text. Is it the case that goal-directed mice in RI group (excluded dots at bottom) exhibited physiological characteristics that were similar to RR group, or alternatively naive mice? How are the cellular physiology results changed if those animals were included? In a previous publication from the group, they combined all mice, and formed a continuous distribution of goal-vs-habit mice.

The experiments to which the reviewer refers are single cell patch clamp experiments, and as such, the technical and biological variation from 1‐2 whole cell recordings/animal simply do not permit the individual subject correlations that the population 2PLSM imaging data enabled. Rather we needed to perform group‐wise comparisons, as is routinely done. In this case, we used the two training protocols that *generally* but don’t completely bias to habit v goal behavior and determined *a priori* that we would take the habitual mice from the habit‐biased protocol and the goal‐directed subjects from the goal‐biased protocol.

We would also like to assure the reviewer that there was no selective data exclusion. The process for determining subjects to study was as follows: the experimenter (JOH) was given trained mice, blind to behavior and training schedule, by another scientist (EG). π was given the LPr data from the probe trials conducted by EG, she (NC) calculated the Normalized Devalued Lever Press rates and then identified the subject numbers along with group assignments that were to be used for recordings to JOH.

If the reviewer wishes, the authors will include a note regarding this nuance in the Materials and methods section.

[Editors' note: the author responses to the re-review follow.]

The manuscript has been improved but there are some remaining issues that need to be addressed before acceptance, as outlined below:1) In this revision the authors have made several essential clarifications and performed additional control experiments and additional analyses, which have strengthened the study in important respects. Despite these improvements, several experiments in my mind are still missing the correct analyses or controls; details follow below. As before, I am enthusiastic about the integrative approach taken by the authors in combining and synthesizing across multiple experimental approaches. However, I detect an overall tendency to constrain the interpretation of specific experiments by referring to an overall synthesis or working model derived from multiple experiments. As I elaborate on below, for me to have more confidence in the results, I expect the authors to first carefully interpret the results from specific individual experiments in the Results section, without reference to an overall favored working model. Then, they are encouraged (in the Discussion) to come up with a synthesis across experiments. What I think they should avoid is writing parts of the results as if this synthesis has already been shown to be correct. This overall issue is illustrated below, particularly in my response to points 3a and 3b.

We thank the reviewer for bringing this stylistic concern to our attention. We have further modified the manuscript to minimize this. To facilitate a quick re-review, below we extracted the main introductory and conclusion sentences throughout the results, and indicated our edits.

Conclusion unmodified:

“These pharmacological experiments in acute brain slices indicate that IEM-1460 promotes an indirect pathway timing advantage and selectively diminishes multi-action potential evoked SPN responses.”

Conclusion added:

“This result confirms that IEM-1460, which inhibits FSI firing (Figure 1—figure supplement 1), selectively reduces multi-action potential SPN responses to afferent stimulation (Figure 1—figure supplement 2) as suggested by calcium imaging experiments (Figure 1).”

Unmodified. In our view, this is an important point to discuss at this point in the paper because the results suggest that the sign of the hypothesis is inverted. In this case, we believe that some reference to the overall working hypothesis is necessary to communicate rationale behind subsequent experiments.

“Altogether, this series of experiments identifies a pharmacological agent that potently inhibits FSI activity and modulates all of the habit-predictive SPN firing properties. These results were surprising for two reasons. First, rather than a blockade of FSI activity causing disinhibition of SPNs as we had hypothesized, we found that when FSI activity was reduced, SPN activity was also reduced. This result suggests that FSI activity is capable of promoting, rather than inhibiting, SPN activity at least in the acute brain slice preparation. Secondly, although IEM-1460 strikingly affected the same features of DLS output that predict the expression of habitual behavior (calcium transient amplitude in both pathways and relative pathway timing) (Figure 1), the directionality of these effects was opposite in all measures. Therefore, these results revise the overall hypothesis to involve a gain, rather than loss, of FSI activity as a candidate mechanism for habitual behavior.”

Unmodified:

“While IEM-1460 has been shown to have selective effects on the firing of FSIs in striatum, its effect of inhibiting AMPAR-mediated excitatory postsynaptic currents (EPSCs) in cholinergic interneurons (CINs) leaves open the possibility that CINs might contribute to our observed IEM-1460 effects.”

Modified. Original phrasing could have been misinterpreted to mean behavior, whereas we meant to refer only to the physiological “habit-predictive” findings at this point.

“Consistent with the IEM-1460 results in 2PLSM calcium imaging (Figure 1) and cell-attached recording (Figure 1—figure supplement 2) experiments, this optogenetic result indicates that FSIs promote multi-action potential SPN responses to cortical excitation in the brain slice and that the effects of IEM-1460 on striatal output occur primarily through a reduction of striatal FSI activity.”

Original:

“This optogenetic result is consistent with the IEM-1460 results in 2PLSM calcium imaging (Figure 1) and cell-attached recording (Figure 1—figure supplement 2) experiments. Taken together, these data indicate that FSIs promote multi-action potential SPN responses to cortical excitation in the brain slice and that the habit-opposing circuit output effects of IEM-1460 occur primarily through a reduction of striatal FSI activity.”

Modified. Although we prefer to remind reader of hypothesis and expectations, we have deleted this part to satisfy the reviewer’s stylistic concerns.

“While results thus far show that FSIs appear capable of specifically modulating habit-predictive properties of striatal output, we next examined whether FSI activity was different as a result of experience. We measured FSI synaptic and cellular electrophysiological properties in DLS brain slices prepared from habitual and goal-directed mice.”

Original:

“While results thus far show that FSIs appear capable of specifically modulating habit-predictive properties of striatal output, FSIs themselves would presumably need to undergo experience-dependent plasticity during the course of habit formation in order to alter their modulation of SPN firing and drive expression of habitual behavior. Because inhibiting striatal FSIs produced circuit effects opposite to those of habit expression, we hypothesized that FSIs would become strengthened with habit formation, for example through increases in synaptic strength and/or excitability. To test this hypothesis, FSI synaptic and cellular electrophysiological properties were measured in DLS brain slices prepared from habitual and goal-directed mice.”

Opening and closing, unmodified:

“Habitual behavior was associated with increased FSI firing in response to somatic current injection. However, it was afferent activation that initially revealed habit-predictive striatal output properties^7^. Therefore, in order for FSI plasticity to alter striatal output, it must be sufficient to differentially drive FSI firing in response to similar coincident synaptic excitation.”

“Together, these experiments show that FSIs undergo long-lasting, experience-dependent plasticity with habit formation and that this plasticity is sufficient to increase FSI firing.”

Modified:

“Since photo-inhibiting FSIs produces striatal output properties that directly oppose those seen in habit (Figure 1), we inhibited FSIs after habit training to determine the necessity of FSI activity for expression of habitual behavior.”

Original:

“Since photoinhibiting FSIs produces striatal output properties that directly oppose those seen in habit (Figure 1), the observed increase in FSI excitability would be predicted to drive the output-level striatal circuit signature for habit. Therefore, to test the necessity of FSI activity for the expression of habitual behavior, mice underwent habit-training protocols in the operant lever press task and then, prior to testing the degree of habitual responding, FSIs were inhibited chemogenetically.”

“These findings show that acute suppression of FSI activity in DLS causes habit-trained subjects to behave as though they were goal-directed.”

Unmodified:

“To understand how chemogenetic suppression of FSI firing affects striatal activity *in vivo*, single unit recordings were performed in a cohort of PV-Cre::Drd1a-tdTomato^26^ mice implanted in DLS with multi-electrode arrays and injected with the Cre-dependent hM4D inhibitory chemogenetic virus.”

Deleted:

“Notably, this trend was reminiscent of the linear relationship between basal calcium transient amplitude and the inhibitory effect of IEM-1460 observed *ex vivo* (Figure 1).”

Final paragraph summarizing *in vivo* FSI experiments, somewhat interpretative, but an important point and immediately precedes Discussion section. We don’t believe this “constrains” thinking to fit our hypothesis, which is the main concern of the reviewer.

“These results demonstrate that FSIs modulate SPN activity in a more complicated manner than previously appreciated. While FSIs can have an overall strongly inhibitory effect *in vivo* on SPN firing as traditionally assumed, we also found evidence that they potentiate activity in a select population of SPNs that displays higher fractions of gamma-frequency spiking. This selective potentiation may be akin to a winner-take-all “focusing” mechanism that increases the signal-to-noise ratio in corticostriatal transmission. According to such a mechanism, the subset of recruited SPNs would be facilitated while the less-relevant, low-gamma SPNs would be suppressed.”

1a) For the order of vehicle and CNO applications, I assumed the worst because there was no mention of session ordering in the original manuscript, and the Figure 1 experiments employed a fixed control followed by treatment order (see my comments on this below). The fact that vehicle/CNO treatments were given on separate days and counterbalanced for order is important information that should be included in the Materials and methods, ideally with the table shown in the rebuttal, so that readers can see that in some cases, there were 10+ weeks between the two experiment days.

Table is included in the Review Response. A sentence describing this explicitly is included in the Materials and methods section of the manuscript.

“For *in vivo* electrophysiological recordings, CNO and vehicle were administered on different days and in counterbalanced order.”

1b) The new inclusion of locomotion data, which show similar distance traveled for both groups, is useful and increases confidence in the interpretation of the DREADD experiments.1c) I apologize for any confusion that my original description may have caused. My point here is quite simple: what is the evidence that relative gamma ISI density, rather than firing rate, is the relevant variable here? In the rebuttal, the authors provide a figure showing a lack of significant correlation between firing rate and the fraction of gamma-band ISIs. But this does not exclude the possibility that if the new Figure 5 would test for a correlation between firing rate (fold change; the quantity plotted on the vertical axis) and baseline firing rate instead of gamma density, the reported correlation would persist. I agree with the points made by the authors that it's entirely possible that low overall firing rates are responsible for the lack of a significant correlation between firing rate and the fraction of gamma-band ISIs, and the Poisson analysis is interesting in its own right (my original suggestion would require a further step: asking if the excess gamma ISI probability predicts firing rate change following CNO). However, these are all indirect points. The most direct way of testing whether invoking gamma ISI density is necessary is to determine if using baseline firing rate yields the same overall result or not. If so, I don't see the need for a more complicated, gamma-ISI related explanation. In the rebuttal, the authors state, "fold change correlated with gamma density, not the rate itself" but I did not see evidence for the latter claim in the manuscript: this needs to be tested and stated explicitly.

We thank the reviewer for this point of clarification, and believe we now address it. As the reviewer suspects, in the analyses below, we do find that it is specifically the “excess gamma probability” that differentiates CNO response from vehicle and not a phenomenon related solely to firing rate. This additional analysis reinforces our original conclusions. We have accordingly expanded on this discussion in the manuscript and in Author response image 5 and Author response image 6 we include the new analyses related to this point.

**Author response image 5. respfig5:** 

A similar analysis using firing rate alone indicates that it is the excess gamma specifically that predicts rate modulation by CNO. We find that, as you would imagine, increased rate increases modulation since this is a statistically improbable event in the setting of low basal firing rates; and accordingly, this effect does not show specificity for CNO over vehicle.

**Author response image 6. respfig6:** 

We have modified the Results section to include these points as follows:

“Since neurons with higher firing rates would be expected to have shorter ISIs in general, we examined the possibility that the fraction of gamma ISIs in SPNs might simply relate to mean firing rate. However, we found that the proportion of gamma-frequency ISIs was unrelated to mean firing rate in baseline single unit SPN recordings before either CNO or vehicle administration (pre-CNO: p = 0.25, n = 23; pre-vehicle: p = 0.28, n = 20). Additionally, we found that SPNs fire significantly more gamma-frequency spikes than expected by Poisson processes matched to firing rate (pre-CNO: t(44) = 5.76, p = 7.67 x 10^-7^, n = 23 SPNs & rate-matched simulations; pre-vehicle: t(38) = 8.24, p = 5.59 x 10^-10^, n = 20 SPNs & rate-matched simulations). Whereas baseline firing rates non-specifically predict fold change in firing rate after both CNO and vehicle injection (CNO: r(22) = -0.61, p = 0.0022, n = 23; vehicle: r(19) = -0.45, p = 0.045, n = 20), the excess probability of gamma-frequency ISIs (observed – expected) specifically predicts rate modulation after CNO (r(22) = -0.52, p = 0.011, n = 23) but not vehicle (r(19) = 0.045, p = 0.85, n = 20). Therefore, gamma-frequency spiking represents a feature of interest in SPNs that predicts whether these output neurons will fire more or less as a consequence of reducing FSI activity.”

2a) Acknowledging the interpretation that the FSI manipulations performed are in effect a disruption of DLS activity, as the authors do now in the Discussion, is appropriate. However, I believe the authors' suggestion that their work is the first to show an effect of DLS manipulation on the expression (rather than acquisition) of habits is overstated. Packard and McGaugh (1996) performed lidocaine injections in the caudate nucleus following acquisition of a response strategy on a plus maze. Post-training baclofen/muscimol infusions into the DLS reduce habitual lever-pressing for alcohol (2012 Biol Psychiatry, 2016 Frontiers Behav Neuro), and has similar effects on lever pressing for cocaine (Zapata et al. J Neurosci 2010). Thus, for different operationalizations of habit and reinforcer, post-training manipulations of DLS have been shown to affect habitual (but not goal-directed) behavior.

We thank the reviewer for pointing these studies out. We very much wish to frame our findings accurately within the field and have modified our conclusions by deleting this point in the manuscript and including the earliest of the above-mentioned citations, copied below:

“…In the present study, by chemogenetically inhibiting PV+ interneurons *in vivo*, we found that FSI activity in DLS is required for the expression of a learned habit (Figure 4); an automated, reward-insensitive behavior quite different from behaviors previously studied. Previous pharmacological inactivation studies have demonstrated a role for DLS in habit expression42, 43, indicating that general disruption of DLS activity also impairs established habitual behavior. Interestingly, in the present study, chemogenetic inhibition of FSI activity drove an overall increase in projection neuron activity (Figure 5) which suggests that reducing FSI activity specifically may impair habit expression differently than a general inactivation of the circuitry.”

2b) Great, inclusion of the lever press and total rewards data strengthens the results.3a) I appreciate the authors are combining multiple approaches and lines of evidence to converge on a proposed mechanism – this is a strength of the paper. But in this case it's hard to see beyond the most direct way to rule out nonspecific changes related to time, which is simply to counterbalance the order of the treatment and control groups, or to collect washout data. The authors second observation provided is the most relevant in that it does contain a treatment vs. control comparison without the potential time confound, but this is a separate experiment, but low n, where we are not told how its timescale maps on to the main experiment, while leaving open whether the effect shown here underlies the full Figure 1 results. The other observations are more indirect still. Fixing this would require re-doing the experiment. However, if the authors feel strongly that they can argue against this confound, this flaw may be outweighed by intriguing results in other parts of the paper that collectively contribute to an emerging new view of FSI-SPN interactions.

For technical reasons, a counter balance or washout design is not possible due to kinetics of drug action in the brain slice (approx. 20 minutes to take effect) and loss of Fura-2 signal with time. We have piloted a second approach to address this reasonable point, in which we perform group-wise comparisons to complement the original within-cell design. Slices are pre-incubated in vehicle or IEM drug prior to recording evoked calcium transients on the 2PLSM.

These new and confirmatory results are now described in the manuscript:

“Because the within-cell experimental design of measuring effects before and after IEM-1460 application did not exclude the possibility that changes in calcium signals occurred during the 20-minute wash-in period independently of IEM-1460, we performed a separate across-group study. Brain slices were incubated with either IEM-1460 or vehicle prior to and during imaging. Group mean calcium transient amplitudes were lower in IEM-1460 relative to vehicle in both dSPNs (vehicle: 0.043 ± 0.0011 ΔF/F0, N = 202 cells; IEM-1460: 0.037 ± 0.0021 ΔF/F0, N = 72 cells; t(272) = 2.62, p = 0.0093) and iSPNs (vehicle: 0.040 ± 0.0014 ΔF/F0, N = 143 cells; IEM-1460: 0.033 ± 0.0011 ΔF/F0, N = 56 cells; t(197) = 2.93, p = 0.0038) and IEM-1460-treated slices showed a preference for faster indirect pathway activation relative to vehicle-treated slices (t(197) = 3.83, p = 1.41 x 10^-7^, N = 143 & 56 independent dSPN/iSPN pairs). These results are generally consistent with findings from within-cell pre-post measurements.”

And corresponding data included here for reviewers’ reference:

**Author response image 7. respfig7:** 

As to the other points, we clarify that the time scale of the vehicle versus IEM-1460 washin experiments in Figure 1—figure supplement 2 is approximately the same as the 2PLSM. Stable cell-attached recordings are obtained, drug is washed in for 20 minutes and then the post condition is recorded (approx. 30 min total).

We are unclear about the comment of other observations are more indirect still. We view Figure 2 as the most direct because we use a genetically defined targeting of FSI activity instead of the IEM drug, and because the laser pulsing on and off is temporally interleaved, controlling for that confound the best.

To emphasize the similarities of observations across experimental paradigms, we had re-analyzed the Figure 2 data in the same was as in Figure 1 and the supplement to Figure 1. This figure was included in the original reviewer response.

3b) As with 3a, I note that the authors' style is to make inferences from multiple experiments to converge on an overall explanation. This is admirable and important. However, this does not absolve them of responsibility in accurately describing what can and cannot be concluded from each experiment separately. I do not think it is appropriate to take their proposed overall mechanism and use it as a source of top-down constraints on the initial interpretation of specific, individual experiments. What I want to see is careful descriptions and conclusions for each individual experiment in the Results that appropriately take the limitations of each into account. Then, the authors are encouraged to integrate the results into an overall proposed mechanism, and if they wish, use the resulting synthesis to inform interpretation of specific pieces of data – but not where those experiments and data are first described.

As above, we reviewed the manuscript for this stylistic concern and believe we have achieved this goal.

For this particular point, while it may be true that about 1% of striatal neurons are FSIs, they tend to be highly active, rendering them more likely to be included in recording studies (and, depending on how cells are identified in imaging studies, in those too; see e.g. Harris et al. Nat Nat Neurosci 2016). I agree with the authors that results such as Figure 1 are unlikely to result from inadvertent inclusion of a few FSIs because we are shown the full distribution of data points. However for others such as Figure 1 which simply compare means, it seems possible that a few potential FSI outliers could shift the means significantly.

We understand the reviewer’s continued concern and include new data explicitly testing the degree to which FSI activity is captured by our activity imaging approach. We conclude that it is not possible for unexpected contamination of “SPN” signal by FSIs to underlie any of our conclusions.

Using a Pv-Cre x Ai9 reporter mouse we monitored calcium transients in genetically identified FSIs. We found that, using our Fura cell loading and data analysis conditions, evoked firing activity was detected in 50% of the total transgenically reported FSIs. We further found that while increasing stimulation intensity can increase FSI spike probability, we did not see a significant increase of the calcium transient amplitude. This observation is consistent with our unpublished observations that FSIs rarely burst fire under our electrical afferent stimulation and recording conditions. Both of these empiric observations indicate that FSI contamination is not a source of our present study’s main findings (i.e. we find modulation of amplitude but not spike probability). We also looked for statistical outliers and did not find support that it was driving mean effects.

**Author response image 8. respfig8:** 

Lastly, we have previously documented the sensitivity and specificity of the Drd1a-tdTomato reporter line and find that Pv-positive cells are not present in td-Tomato expressing cells (Ade et al., Frontiers, 2011, Figure 3) which specifically indicates that FSI contamination could not explain any findings in the putative dSPN population.

To detect outliers, we employed the Robust regression and Outlier removal (ROUT) method available in the GraphPad Prism software. This outlier detection analysis fits a robust nonlinear regression to the data and detects outliers by residuals between the data and the robust fit based on a preset false discovery rate (we used Q = 1%).

We detected zero outliers in our pre-IEM iSPN data set, indicating that a small fraction of misclassified cells does not drive our finding:

iSPNs ACSFiSPNs IEM-1460MethodROUT (Q = 1.000%)Number of pointsAnalyzed5252Outliers00

In case a subset of cells showed an outlying response to IEM-1460 rather than an outlying baseline latency, we also ran this analysis for absolute pre-post latency differences in the iSPN data:

iSPN latency differencesMethodROUT (Q = 1.000%)Number of pointsAnalyzed52Outliers0

3c) This additional explanation is helpful.3d) Great, this new figure fully addresses my concern.4) The comments related to behavior-physiology studies (Figure 3) were circumnavigated. The authors DO report an exclusion criterion, namely; "RRshort-trained mice with NDLPr <0 were considered to be goal-directed…Mice not meeting inclusion criterion were not used…". But, as part of their conclusions they routinely refer to the task as " habit promoting reinforcement protocol", and "habit-trained subjects…". Figure 3—figure supplement 1 indicates that 4 of 9 mice trained in this task do not engage in a habitual behavior, an insignificant 44% of mice.

We have re-read our original reviewer response and do not understand what aspect we “circumnavigated”. We understood the point of the reviewer to be whether the mice in the habit promoting RI protocol that had values indicating goal-oriented behavior had physiological properties that were more goal-like, more habit-like, or other. We clarified by stating that we never recorded from these mice and excluded them *a priori*. We could not evaluate the relationship of physiology across a behavioral continuum in Figure 3 because single cell recording variation is underpowered relative to our population-based calcium transient data presented in O’Hare et al., 2016. Secondly, we did not exclude any mice in Figure 4, the chemogenetic behavior experiment except for those with no detectable viral expression.

We fully agree and believe that we acknowledge in the manuscript that RI training does not produce total insensitivity to outcome devaluation in 100% of subjects. It is for this reason that we refer to it as a “habit *promoting*” protocol rather than “determining”, for example.

In Figure 4, the authors exposed mice to the behavioral protocol that already produces ~44% of goal directed behavior, to argue that DREADD manipulation produces robust goal directed behavior.So, without a devaluation test before CNO day to assess degree of habit induced, it is hard to accept the claim that CNO promoted less habitual behavior. If the physiological phenotype of the intermediate animals (trained on task, but did not exhibit habitual behavior) were known, I would be more convinced that CNO inactivation of FSI would (at least) shape an incomplete phenotype described in Figure 3 really thing Figure 4 has compelling results, and hope the authors address this concern.

This is a very interesting point and one that we can agree with in principle but not totality. The 9 mice cohort in Figure 3 is not appropriate to generalize about the effectiveness of RI reinforcement in driving habitual behavior. To better determine the effects of RI reinforcement, we have now analyzed the data from every mouse that we have trained on this reinforcement schedule (N = 59) and found that RI-trained mice display a mean NDLP_r_ of 0.052 ± 0.13.

More importantly, behavioral results vary between cohorts for a number of reasons having to do with animal housing conditions and subtle differences in training conditions and trainer (see Sorge, et al. 2014 Nature Methods for a recent example).

**Author response image 9. respfig9:** 

The variance of the data, from 59 RI-trained mice across 8 different experiments, is 1.06. However, when variance is measured within-group, variance is significantly lower (0.86), consistent with our observations that within training cohort are the best comparators. In addition, the aforementioned concern of the poor rate of instantiating habitual behavior refers to one of our weakest habit-inducing cohorts. For these reasons, we remain convinced that it is most appropriate to compare an experimental group to a contemporaneous control group rather than one produced separately in time, as we have done in this study.

Nevertheless, for the reviewer’s benefit, even when we compare the PV-hM4D mice to all 59 historical data points, we see a nearly significant difference whereas control PV-eYFP mice are more similar to unmodified RI-trained mice:

**Author response image 10. respfig10:** 

But as it stands now, I am not fully convinced that the hm4D manipulation produces less habitual behavior. The author could come up with a clever, but robust statistical approach to assess/argue more convincing statistical result. For example if they serially excluded 40% of subjects in both groups, and performed the test, what fraction of the comparisons are significant? Or another formulation of shuffle tests. As I pointed out, a within-subject experiment would have been most ideal; that is each animal (in both PV-hM4D and OV-GFP groups) getting two devaluation sessions with and without CNO. If the authors cannot address this weakness via reanalysis of existing data, I think they will have to demonstrate a more convincing behavioral result with a new cohort of mice.

We would like to note that the study in Figure 4 is powered to the standard 1-β = 0.80 and α = 0.05 parameters for effect detection. For us to remove data and rerun the analysis would be to remove power from our study, which does not seem appropriate.

However, we appreciate the reviewer’s point that behavioral outcomes can be variable and that we may have “gotten lucky” (the p-value suggests a 1.6% chance of this happening), so we ran the clever shuffling analysis suggested by the reviewer based on the success rate of RI reinforcement driving habitual behavior in the control cohort (73% success). We serially removed random data points from both experimental and control groups, repeating this shuffling test N_hm4D_ x N_eYFP_, or 110 times. In Author response image 11 is the distribution of resulting p-values from two-tailed, unpaired t-tests:

**Author response image 11. respfig11:** 

Our mean p-value from this shuffling experiment was also within the 95% confidence range (p = 0.048).To promote confidence that our analysis was carried out as suggested, MATLAB code is pasted below for reviewers’ reference:

function p = leaveNout(exp,ctrl)

groups = {'exp','ctrl'};

threshold = 0;

Ns = [length(exp) length(ctrl)];

fracRemove = sum(ctrl < threshold)/length(ctrl);

numRemove = floor(fracRemove.*Ns);

numShuffles = prod(Ns);

p = nan(numShuffles,1);

for i = 1:numShuffles

for g = 1:length(groups)

removeInds = randperm(Ns(g),numRemove(g));

eval(sprintf('curr_%s =% s;', groups{g}, groups{g}))

eval(sprintf('curr_%s(removeInds) = [];', groups{g}))

end

[~, p(i)] = ttest2(curr_exp, curr_ctrl);

end

[Editors' note: further revisions were requested prior to acceptance, as described below.]Thank you for resubmitting your work entitled "Striatal fast-spiking interneurons drive habitual behavior" for further consideration at eLife. Your revised article has been favorably evaluated by a Senior editor, a Reviewing editor (Michael Frank), and one reviewer.We have sent the manuscript back to the one reviewer who had more reservations (the other reviewer was largely satisfied in the last round and we determined that you have addressed their remaining issues). The manuscript has been improved but there are now just a few remaining issues that need to be addressed before acceptance, as outlined by the reviewer below:This resubmission contains several additional control experiments and analyses, which collectively address three major issues identified as remaining after the last round of reviews, with mixed success. One case is convincing, the second exacerbates my concerns about the authors' interpretation, and the third appears reasonable but is not incorporated into the manuscript.– My request to control for the effects of time in the IEM-4160 slice experiments (Figure 1) is fully satisfied by the addition of new, across-group experiments for slices pre-incubated with IEM and vehicle.– The authors claim that gamma-band ISIs are predictive of firing rate changes in SPN firing following CNO. However, pre-CNO firing rate is a numerically better predictor of such changes than excess gamma-band ISIs (r = -0.61 vs -0.45), casting doubt on the relevance of this feature given that a simpler and more predictive measure is available. The authors argue that firing rate "non-specifically" predicts such changes (i.e. for both vehicle and CNO) and that therefore excess gamma-band ISIs (which only predict post-CNO firing rates, but not post-vehicle rates) is an interesting quantity; however, these results could also be interpreted to mean that gamma-band ISIs is simply a worse predictor overall. I could imagine additional analyses, such as a multiple regression, that could help clarify this issue; however, regardless of the specific outcome, any remaining variance explained will likely be low and of unclear relevance. In addition, the authors should be aware that a comparison between two t-tests (one significant, one non-significant) does not constitute sufficient evidence for an interaction (see Nieuwenhuis et al. Nat Neurosci 2011) and report the correct test instead.

Per Nieuwenhuis et al. “When making a comparison between two correlations, researchers should directly contrast the two correlations using an appropriate statistical method”. We would like to first note that this article specifically discusses situations when p-values are quite similar, but fall on each side of 0.05 (such as 0.05 and 0.06) and the sign of the observed effects are similar. This is not our particular situation. Our p values for predicting CNO versus vehicle modulation are 0.011 and 0.85, respectively.

In any case, we now include this additional analysis. We performed a Fisher r-to-z transformation to assess the significance of the difference between two correlation coefficients. The results are detailed below, but in summary, these analyses show that our main conclusions are supported and remove this remaining concern.

Using the Fisher r-to-z transformation we demonstrate three points related to the reviewer’s concerns.

1) Firing rate is not a “better predictor” than excess gamma band ISIs for the effects of CNO on rate modulation. The two r-values (r = -0.61 vs. -0.52) are not significantly different, p = 0.34, z = -0.42.

As a reminder of our values used to perform these calculations:

Correlation of CNO modulation with baseline firing rate: r = -0.61, p = 0.0022, n = 22.

Correlation of CNO modulation with baseline excess gamma: r = -0.52, p = 0.011, n = 22.

(The reviewer mistakenly cites -0.61 v -0.45, which are the correlation values for baseline firing rate vs rate modulation by CNO and VEH groups. These two correlations are also not significantly different from one another: p = 0.25, z = -0.68.)

2) By contrast, the correlation coefficients for baseline excess gamma predicting rate modulation DO significantly differ between VEH and CNO,

p = 0.030, z = -1.88.

Based on our findings of:

Correlation for CNO: r = -0.520, p = 0.011, n = 23

Correlation for VEH: r = 0.045, p = 0.850, n = 20

We now report these measures in revised manuscript:

“Since neurons with higher firing rates would be expected to have shorter ISIs in general, we examined the possibility that the fraction of gamma ISIs in SPNs might simply relate to mean firing rate. […] Therefore, gamma-frequency spiking represents a feature of interest in SPNs that predicts whether these output neurons will fire more or less as a consequence of reducing FSI activity.“

In summary, we adopted an appropriate statistical test for comparing correlational analyses and found that baseline excess gamma activity indeed specifically predicts how the firing rates of individual SPNs will be modulated by chemogenetic inhibition of FSI activity, in comparison to Vehicle. Secondly, we found that SPN modulation by CNO was not predicted better by baseline firing rates than baseline gamma density. And thirdly, we found that baseline firing rates also do not predict SPN modulation by CNO specifically, but rather baseline firing rate correlates with rate modulation equally well for both CNO and VEH.

(N.B. If interest in this last point remains, we would like to clarify that we believe these are the relationships you would expect to find by chance due to the law of large numbers and a tendency to regress to the mean. We have directly tested this using random samplings of our data shuffled and find that our observed values are within those predicted by the simulation.)

– In order to address the concern that FSIs included in the iSPN dataset (which is defined by the absence of D1R label) may contribute to the effects in Figure 1, the authors provide new data suggesting this is unlikely. At a minimum the authors need to discuss this possible confound in the manuscript, and ideally include the data shown in a supplementary Figure.

As requested, we now include discussion on interneuron contamination of datasets in the manuscript and the 2PLSM FSI data as a new supplementary figure.

“Contamination of dSPN and iSPN datasets by interneurons was minimized by selection criteria and monitoring datasets for outliers (See Materials and methods for further details).”

“Classifying regions of interest: Regions of interest (ROIs) showing red and green were classified as dSPNs whereas green-only ROIs were classified as iSPNs. The small percentage of green-only cells which would have been striatal interneurons was partially mitigated by ignoring abnormally large ROIs which were likely to be cholinergic interneurons. FSIs comprise approximately 1% of all striatal neurons and are not present in the Drd1a-TdTomato labelled population. However, FSIs might be included in the “putative iSPN” population. Empiric experiments using Pv-Cre x Ai9 reporter mice demonstrate that approximately 50% of FSIs are labelled with Fura2-AM and pass data inclusion criteria in our experimental setting (see Materials and methods and Figure 1—figure supplement 1). Therefore, we expect at most that approximately 1% of iSPN and none of dSPN data may represent FSIs. Because CIN and FSI firing properties are very different from SPNs, we reviewed our data sets for the presence of outliers using Robust regression and Outlier (ROUT) analysis (GraphPad Prism software) and a false discovery rate of Q = 1%. No outliers were detected in iSPN baseline amplitude, post-IEM-1460 amplitude, or latency modulation datasets of Figure 1.”